# GRADPOWER: POWERING GRADIENTS FOR FASTER LANGUAGE MODEL PRE-TRAINING

## ABSTRACT

We propose **GradPower**, a lightweight gradient-transformation technique for accelerating language model pre-training. Given a gradient vector $\boldsymbol{g} = (g_i)_i$, GradPower first applies the elementwise `sign-power` transformation:

$$\varphi_p(\boldsymbol{g}) = (\text{sign}(g_i)|g_i|^p)_i$$

for a fixed $p > 0$, and then feeds the transformed gradient into a base optimizer. Notably, GradPower requires only a **single-line code change** and no modifications to the base optimizer's internal logic, including the hyperparameters. When applied to Adam (termed **AdamPower**), GradPower consistently achieves lower terminal loss across diverse architectures (LLaMA, Qwen2MoE), parameter scales (66M to 2B), datasets (C4, OpenWebText), and learning-rate schedules (cosine, warmup-stable-decay). The most pronounced gains are observed when training modern mixture-of-experts models with warmup-stable-decay schedules. GradPower also integrates seamlessly with other state-of-the-art optimizers, such as Muon, yielding further improvements. Finally, we provide theoretical analyses that reveal the underlying mechanism of GradPower and highlights the influence of gradient noise.

## 1 INTRODUCTION

Large language models (LLMs) have revolutionized modern artificial intelligence (Brown et al., 2020; Achiam et al., 2023; Liu et al., 2024a). However, pre-training LLMs is computationally intensive due to the massive scale of model size and training data. Improving the pre-training efficiency has thus become a primary objective in the continued scaling of LLMs. Among the factors affecting efficiency, the choice of optimizer is critical. In practice, the Adam optimizer (Kingma & Ba, 2014; Loshchilov & Hutter, 2017) has emerged as the de facto choice in most LLM pre-training pipelines, owing to its adaptive learning rate features (Zhang et al., 2024; Kunstner et al., 2024).

To further accelerate Adam, several approaches have been proposed to refine or simplify its moment estimation (Xie et al., 2024; Pagliardini et al., 2025; Chen et al., 2024b; Liu et al., 2025b; Zhang et al., 2025). Other strategies modify the update rule directly, such as direction correction (Wang et al., 2024; Liang et al., 2024), incorporating curvature information (Liu et al., 2024b; Wang et al., 2025), or applying matrix-based preconditioning (Keller et al., 2024; Liu et al., 2025a). While these methods often deliver tangible gains, they typically require substantial modifications to the existing training pipeline and careful extra hyperparameter tuning, which hinders their practical adoption.

In contrast to these intrusive modifications, we instead propose a lightweight, plug-in approach by revisiting the core component of optimization: the *gradient itself*. We apply a simple elementwise transformation to the gradient vector – enhancing its informativeness while leaving the base optimizer entirely unchanged. This design preserves compatibility with existing training pipelines and avoids additional tuning burden. Specifically, **our contributions** are as follows:

- **Our approach.** We propose GradPower, a simple but effective approach for boosting the convergence of general gradient-based optimizers. Specifically, given a raw gradient $\boldsymbol{g} = (g_i)_i \in \mathbb{R}^d$ and a fixed $p > 0$, we define the **powered gradient** as

$$\varphi_p(\boldsymbol{g}) := |\boldsymbol{g}|^p \, \text{sign}(\boldsymbol{g}) = (|g_1|^p \, \text{sign}(g_1), \ldots, |g_d|^p \, \text{sign}(g_d))^\top . \tag{1}$$

GradPower applies this powered gradient to the base optimizer, preserving its original structure.

- **Empirical performance.** We first evaluate the effectiveness of GradPower by integrating it into Adam, termed **AdamPower**. We test its performance across a broad LLM pre-training landscape: **dense** models (LLaMA (Touvron et al., 2023)) and **mixture-of-experts** models (Qwen2MoE (Yang et al., 2024a)), ranging from **66M** to **2B** parameters, using the C4 and OpenWebText corpora, and under both cosine-decay (cos) and warmup-stable-decay (wsd) learning-rate schedules. Across all settings, AdamPower consistently achieves lower terminal loss and exhibits more favorable scaling laws compared to vanilla Adam (see Figure 1), demonstrating its potential for improved scalability to larger models. Notably, the performance gains are most significant for modern MoE architectures and wsd schedules.

  Furthermore, we show that GradPower can be also seamlessly integrated with other state-of-the-art optimizers, such as Muon (Keller et al., 2024; Liu et al., 2025a) and Blockwise LR (Wang et al., 2025), yielding additional performance improvements.

- **Theorectical analysis.** (1) Recent analyses suggest that steady progress along flat "river-like" directions is crucial for reducing loss in LLM pre-training (Wang et al., 2024; Wen et al., 2025). We show that AdamPower amplifies these directions, thereby accelerating optimization. (2) Moreover, for general *smooth non-convex objectives*, we prove that augmenting adaptive optimizers (e.g., AdaGrad) with GradPower strictly accelerates their convergence in both low- and high-noise regimes, supporting the intuitions developed in Section 2.

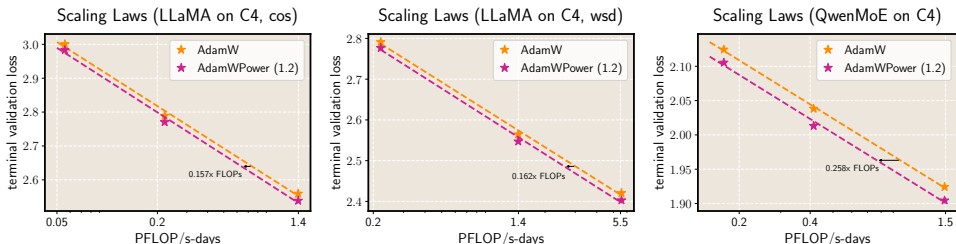

Figure 1: Scaling-law comparison of AdamPower and Adam on the C4 dataset for dense LLaMA models and mixture-of-experts Qwen2MoE models.

**Notations.** For $\{a_s\}_{s=1}^{\infty}$, its $\beta$-exponential moving average at time $t$ is denoted as $\texttt{EMA}_\beta(\{a_s\}_1^t) := (1-\beta)\sum_{s=1}^{t}\beta^{t-s}a_s$. For $g \in \mathbb{R}$ and $p \in \mathbb{R}_+$, we denote $g^p := |g|^p \operatorname{sign}(g)$; for a vector $\boldsymbol{g}$, the notation $\boldsymbol{g}^p$ denotes element-wise application of this transformation. For simplicity, we use *a.s.* to denote "almost surely", and use *w.r.t.* to denote "with respect to". We use standard big-O notations $\mathcal{O}(\cdot), \Omega(\cdot), \Theta(\cdot)$ to hide problem-independent constants, and use $o(\cdot)$ to denote the infinitesimal. Let $\|\cdot\|_q$ denote the $\ell_q$ norm for vectors for a $q > 0$. We denote $[n] = \{1, \cdots, n\}$ for an integer $n \in \mathbb{N}_+$.

## 2 THE GRADPOWER APPROACH

Let $\boldsymbol{g}_t \in \mathbb{R}^d$ denote the stochastic gradient at step $t$. A gradient-based optimizer can be expressed as $\boldsymbol{\theta}_{t+1} = \boldsymbol{\theta}_t - \eta_t \mathcal{Q}(\boldsymbol{g}_1, \cdots, \boldsymbol{g}_t)$, where $\eta_t$ is learning rate and $\mathcal{Q}$ denotes update rule.

**A unified view of preconditioning.** In practice, raw gradients may not be sufficiently informative or stable, and thus, it is common to transform the gradients before applying the update rule. This leads to the general form of *preconditioned optimizers*:

$$\boldsymbol{\theta}_{t+1} = \boldsymbol{\theta}_t - \eta_t \mathcal{Q}\left(\varphi(\boldsymbol{g}_1), \cdots, \varphi(\boldsymbol{g}_t)\right) \tag{2}$$

where $\varphi$ denotes a transformation (or preconditioning) applied to each gradient.

To *avoid computational overhead*, we restrict attention to *elementwise* transformations. For a gradient vector $\boldsymbol{g} = (g_i)_i \in \mathbb{R}^d$, we consider $\varphi(\boldsymbol{g}) := (\varphi(g_1), \ldots, \varphi(g_d))^\top \in \mathbb{R}^d$, where $\varphi : \mathbb{R} \to \mathbb{R}$ is a scalar nonlinearity applied coordinate-wise. The function $\varphi$ is designed to enhance the informativeness of the raw gradient. The simplest choice is a linear transformation $\varphi(z) = cz$ with $c \in \mathbb{R}$. However, as shown in Appendix C, such linear preconditioners may exhibit limited effectiveness when used in modern optimizers for LLM pre-training. In this work, we explore **nonlinear preconditioning** as an alternative.

**The GradPower family.** Empirically, we find that a simple power transformation already yields nontrivial improvements in LLM pre-training. Specifically, we define the sign-power transformation

$\varphi_p : \mathbb{R} \to \mathbb{R}$ with exponent $p > 0$ as

$$\varphi_p(z) = |z|^p \text{sign}(z).$$

The powered gradient is shown in Equation (1). Incorporating this transformation into Adam leads to a new optimizer we call **AdamPower**, detailed in Algorithm 1. Remarkably, AdamPower introduces only one additional line of code compared to standard Adam.

In the following sections, we first present empirical evidence demonstrating the effectiveness of AdamPower, followed by a theoretical analysis that sheds light on its underlying mechanisms.

---

**Algorithm 1: AdamPower** (with decoupled weight decay)

---

**Given** learning rates $\{\eta_t\}_{t=1}^T$; hyperparameters $\beta_1, \beta_2, \epsilon, \lambda$; power $p \in \mathbb{R}_+$;

**Initialize** $\boldsymbol{\theta}_0 \in \mathbb{R}^d$, first momentum vector $\boldsymbol{m}_t \leftarrow \boldsymbol{0}$, second momentum vector $\boldsymbol{v}_t \leftarrow \boldsymbol{0}$;

**for** $t = 1, \cdots, T$ **do**

    compute the mini-batch gradient $\boldsymbol{g}_t$;

    **GradPower**: compute powered gradient $\boldsymbol{g}_t \leftarrow |\boldsymbol{g}_t|^p \text{sign}(\boldsymbol{g}_t)$ using Eq. (1);

    $\boldsymbol{m}_t \leftarrow \beta_1 \boldsymbol{m}_{t-1} + (1 - \beta_1)\boldsymbol{g}_t$;    $\hat{\boldsymbol{m}}_t \leftarrow \boldsymbol{m}_t/(1 - \beta_1^t)$;

    $\boldsymbol{v}_t \leftarrow \beta_2 \boldsymbol{v}_{t-1} + (1 - \beta_2)\boldsymbol{g}_t^2$;    $\hat{\boldsymbol{v}}_t \leftarrow \boldsymbol{v}_t/(1 - \beta_2^t)$;

    $\boldsymbol{\theta}_t \leftarrow \boldsymbol{\theta}_{t-1} - \eta_t \left( \hat{\boldsymbol{m}}_t / \left( \sqrt{\hat{\boldsymbol{v}}_t} + \epsilon \right) + \lambda \boldsymbol{\theta}_{t-1} \right)$;

**Output:** optimized parameters $\boldsymbol{\theta}_T$.

---

## 3 EMPIRICAL EVALUATION

### 3.1 EXPERIMENTAL SETUP

We evaluate AdamPower for the task of LLM pre-training across a range of model architectures, parameter scales, datasets, and learning rate (LR) schedulers. The main experimental configurations are summarized below, while additional implementation details are provided in Appendix B.

- **Models.** We consider two widely used LLM architectures: **LLaMA** (dense) models (Touvron et al., 2023) and **Qwen2MoE** (MoE) models (Yang et al., 2024a). We experiment with model sizes ranging **from 66M to 2B** parameters.

- **Datasets.** We evaluate our methods on the **Colossal Clean Crawled Corpus (C4)** dataset (Raffel et al., 2020)[1] and **OpenWebText** dataset (Gokaslan & Cohen, 2019)[2]. For pre-training on C4, we follow the setup of Wortsman et al. (2024); Zhao et al. (2025), using a batch size of 512. The total number of training tokens is set to be approximately 20 times the number of model parameters, in accordance with the Chinchilla scaling law (Hoffmann et al., 2022).

- **LR schedulers.** We evaluate two popular LR scheduling strategies: (i) cos: a linear warm-up to peak lr_max, followed by cosine decay to a terminal LR lr_min. (ii) wsd (warmup-stable-decay scheduler) (Zhai et al., 2022; Hu et al., 2024; Hägele et al., 2024): a linear warm-up LR to peak lr_max, followed by a stable phase where LR remains at lr_max (up to 80% of the total training steps), and then a linear decay to lr_min. It should be noticed that wsd introduces a non-traditional loss curve: slowly decrease during the stable phase and suddenly drop during the final decay phase.

We further evaluate our method on vision tasks, and report detailed implementation settings in Appendix B.

**Adam Baselines.** We use the standard Adam optimizer (with decoupled weight decay) as the baseline in most experiments (expect Section 3.4). The baseline is configured with hyperparameters $\beta_1 = 0.9, \beta_2 = 0.95$, weight decay $\lambda = 0.1$, and gradient clipping threshold of 1.0, following protocols used in LLaMA pre-training (Touvron et al., 2023). For each experiment, we first tune lr_max over {1e-4, 2e-4, 3e-4, 6e-4, 1e-3, 1.5e-3} to be optimal for Adam, and the baselines are trained using these optimal lr_max's. Details can be found in Appendix B.

---

[1] A large-scale public language datasets, widely used for LLM pre-training such as T5 (Raffel et al., 2020), and prior pre-training studies (Zhao et al., 2024; 2025).

[2] An opensource recreation of the WebText corpus, commonly used in pre-training models such as RoBERTa (Liu et al., 2019), GPT-2, and NanoGPT (Karpathy, 2022).

**The tuning of power $p$ and its transferability.** We only tune the power $p$ in a single small-scale experiment: pre-training LLaMA (0.2B) on C4. As shown in Figure 2, the tuned power is 1.2. Interestingly, this aligns with the optimal power observed in the high-noise regime of our illustrative toy example (Figure 7). Then, we adopt $p = 1.2$ as the default in most experiments (expect Section 3.5). Importantly, the power proves to be **highly robust**: AdamPower with $p = 1.2$ **consistently outperforms** Adam and exhibits better scaling laws, across model types, model sizes, datasets, and LR schedulers.

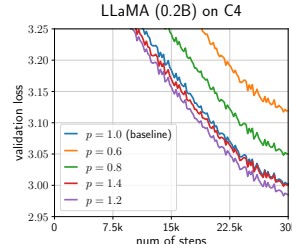

Figure 2: Pre-training LLaMA (0.2B) on C4 using AdamPower with different power $p$'s. The optimal power is 1.2.

## 3.2 RESULTS ON DENSE MODELS

**Main findings.** Figure 3 compares the performance of AdamPower (with $p = 1.2$) to that of vanilla Adam across a range of settings, including LLaMA models of size 66M, 0.2B, 0.4B, 1B and 2B; both `cos` and `wsd` LR schedulers; and the C4 and OpenWebText datasets. Across all experiments, AdamPower **consistently achieves a lower terminal loss** than well-tuned Adam baseline. To further assess its scalability, we visualize the **scaling laws** of AdamPower versus Adam in Figure 1 (left and middle). We observe that the performance gain of AdamPower over Adam remains consistent across a wide range of model scales, **highlighting the potential scalability of AdamPower**.

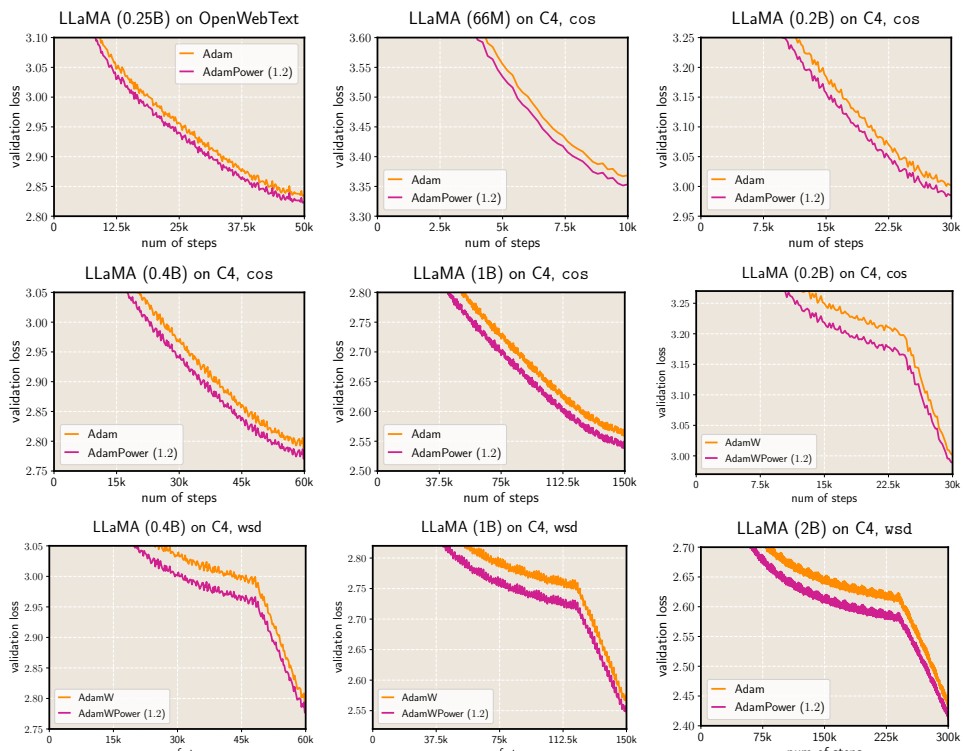

Figure 3: AdamPower ($p = 1.2$) consistently outperforms Adam in LLaMA pre-training tasks across a range of model sizes, datasets and LR schedulers.

**Evaluation on downstream tasks.** Additionally, We also evaluate zero-shot performances of our method on common benchmarks including ARC (Yadav et al., 2019), PIQA (Bisk et al., 2020), HellaSwag (Zellers et al., 2019), OBQA (Mihaylov et al., 2018), WinoGrande (Sakaguchi et al., 2021), using the lm-evaluation-harness codebase (Gao et al., 2024). The results are reported in in Table 1. The model pre-trained with AdamPower outperforms that trained with AdamW on five out of six tasks, as well as on the overall average score, demonstrating improved downstream performance under the same number of pre-training steps.

| METHOD | ARC-E | ARC-C | PIQA | HELLASWAG | OBQA | WINOGRANDE | AVG. |
|---|---|---|---|---|---|---|---|
| AdamW | 60.02 | **26.45** | 73.56 | 44.65 | 24.80 | 56.83 | 47.72 |
| AdamPower (1.2) | **60.35** | 26.28 | **73.61** | **44.93** | **25.00** | **59.43** | **48.26** |

Table 1: The evaluation results of LLaMA (2B) models pre-trained using the C4 dataset. The best scores in each column are bolded.

### 3.3 RESULTS ON MoE MODELS

Mixture-of-experts (MoE) architectures have emerged as a key design choice in modern LLMs, as exemplified by Qwen-2.5 (Yang et al., 2024b) and DeepSeek-V3 (Liu et al., 2024a). Compared to dense models, MoE models often exhibit greater training instability. To assess whether the benefits of AdamPower extend to MoE models, we conduct experiments on Qwen2MoE (Yang et al., 2024a).

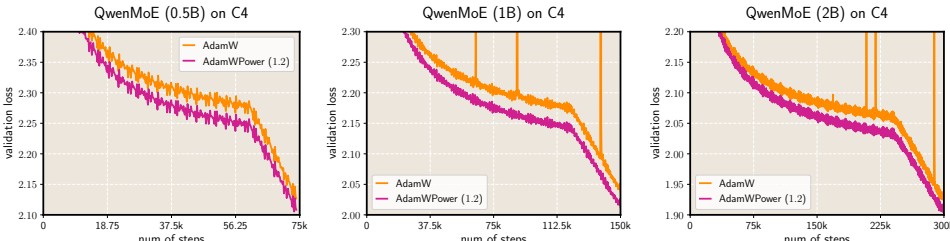

Figure 4: AdamPower ($p = 1.2$) consistently outperforms Adam in QwenMoE pre-training tasks on C4, across varying model sizes. The learning rate schedule is `wsd`.

**Main findings.** Figure 4 compares the performance of AdamPower ($p = 1.2$) and standard Adam for pre-training QwenMoE models of sizes 0.5B, 1B, and 2B on the C4 dataset, using the `wsd` scheduler. Across all settings, AdamPower **consistently achieves a lower terminal loss** than the well-tuned Adam baseline. To further examine scaling behavior, Figure 1 (right) visualizes the **scaling laws** of AdamPower versus Adam during Qwen2MoE pre-training. The performance gap between the two optimizers remains stable across model scales, with the corresponding scaling curves remaining nearly parallel – *suggesting that the gains offered by AdamPower may persist at larger model scales*.

**Special potential for MoE models.** Additionally, we observe two surprising phenomena, suggesting that AdamPower may offer unique advantages for MoE model training:

- Although the power $p = 1.2$ was originally tuned for LLaMA, it generalizes well to Qwen2MoE models without further tuning. (it is likely that an even better $p$ exists for MoE-specific training.) Remarkably, the absolute improvement achieved by AdamPower on Qwen2MoE-2B (0.028) is **more significant** than that on LLaMA-2B (0.022). Noteworthily, Qwen2MoE-2B reaches a much lower loss (1.93) compared to LLaMA-2B (2.43), making further improvements more challenging – yet AdamPower still yields remarkable gains.

- AdamPower also exhibits improved **training stability**, reducing the occurrence of loss spikes seen with Adam. This effect is particularly visible in the 1B and 2B curves in Figure 4 (middle, right). Based on recent understanding in Section A, the fast vibrations along the sharp (valley) directions mainly decide the training (in)stability. We *hypothesize* that the gradient power transformation in AdamPower may help suppress the vibrations along these directions. We leave a detailed investigation of this phenomenon to future work.

- The `wsd` scheduler has become increasingly popular in recent LLM pre-training (Liu et al., 2024a; Hägele et al., 2024), always taking a long stable phase. We observe that the advantage of AdamPower **gradually increases** *throughout the LR stable phase*. This suggests that AdamPower may be particularly suited for modern training pipelines that adopt `wsd` schedules.

### 3.4 COMPATIBILITY WITH OTHER OPTIMIZERS

As discussed in Section A, several optimizers have recently been proposed to enhance LLM pre-training. While AdamPower has demonstrated superiority over Adam in both dense and MoE models, we now ask: *can GradPower also improve the performance of other state-of-the-art optimizers?*

To investigate this, we focus on two representative optimizers: Adam with **Blockwise LR** (Wang et al., 2025) and **Muon optimizer** (Keller et al., 2024; Liu et al., 2025a). Blockwise LR assigns

separate learning rates to different Transformer blocks and has shown substantial improvements over standard Adam. Muon, on the other hand, breaks away from the Adam framework entirely and has recently been shown to achieve better scaling laws than Adam (Liu et al., 2025a). We refer to the application of GradPower to Muon as **MuonPower**.

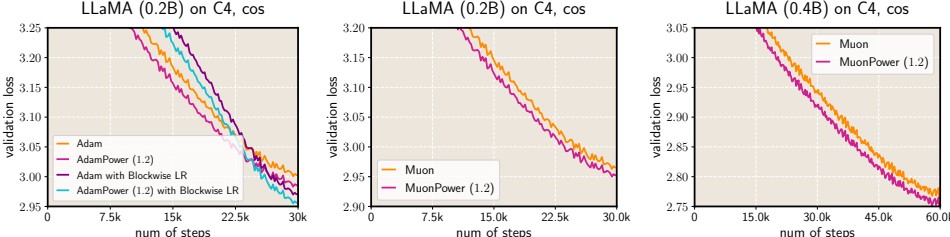

Figure 5: (left) AdamPower with Blockwise LR outperforms both AdamPower and Adam with Blockwise LR in LLaMA pre-training. (middle, right) MuonPower (with $p = 1.2$) outperforms Muon in LLaMA pre-training.

The results, presented in Figure 5, highlight two key findings. **(i) AdamPower with Blockwise LR** achieves a lower terminal loss than both AdamPower and Adam with Blockwise LR individually. Notably, the observed improvement $(0.45)$ is *nearly the sum* of the gains from AdamPower alone $(0.15)$ and Blockwise LR alone $(0.3)$, suggesting that their benefits are largely orthogonal. **(ii) MuonPower** $(p = 1.2)$, the GradPower-augmented variant of Muon, also outperforms the well-tuned Muon baseline. These results demonstrate the versatility of GradPower as a general enhancement that can be seamlessly integrated into other optimizers.

## 3.5 INFLUENCE OF BATCH SIZE

Finally, we investigate how batch size influence the performance of GradPower. Batch size plays a critical role in deep learning, with larger batch sizes producing lower gradient noise and more accurate gradient (Keskar et al., 2017; McCandlish et al., 2018).

Unlike the previous experimental settings, here we conduct the experiments on C4 dataset, varying the batch size from the standard 512 up to 8192. For each batch size, we evaluate AdamPower with multiple values of $p$, and record their validation loss of when the optimal validation loss reaches approximately 3.5. The experimental details are provided in Appendix B.

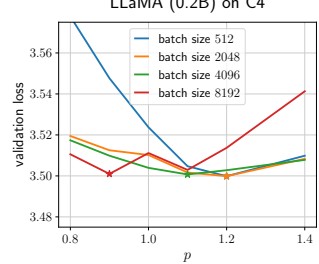

Figure 6: The influence of batch size for the optimal power $p$ in LLM tasks.

**Main findings.** The results, shown in Figure 6, demonstrate a clear trend: the optimal power $p$ decreases as batch size increases, i.e., as the gradient noise level decreases. This finding reveals a strong correlation between batch size and the optimal power $p$ in AdamPower. For standard (small) batch sizes, the optimal power $p$ tends to be greater than 1; in contrast, for large batch sizes, the optimal power $p$ might fall below 1.

**Vision tasks.** We also conduct the experiments using ResNet-34 model (He et al., 2016) on CIFAR-10 dataset (Krizhevsky & Hinton, 2009), varying the batch size from 32 to 128. The results in Table 2 further validates above point. Morever, it demonstrates the generalizability of our method beyond language model pre-training.

| batch size | $p = 0.8$ | $p = 0.85$ | $p = 0.9$ | $p = 1.0$ | $p = 1.1$ | $p = 1.2$ | $p = 1.4$ |
|---|---|---|---|---|---|---|---|
| 128 | **94.35** | 94.27 | 94.22 | 93.98 | 93.38 | 93.15 | 91.66 |
| 64 | 94.22 | **94.4** | 94.22 | 94.1 | 93.97 | 93.77 | 92.61 |
| 32 | 94.04 | 94.07 | 94.15 | **94.3** | 94.25 | 93.85 | 93.71 |

Table 2: The influence of batch size for the optimal power $p$ in vision tasks.

In the next section, we provide a theoretical explanation for this phenomenon.

# 4 THEORETICAL INSIGHTS

## 4.1 AN ILLUSTRATIVE CASE STUDY

This subsection investigates a phenomenological example, both theoretically and empirically, to illustrate how varying the power $p$ in AdamPower affects the update magnitude. Motivated by the empirical findings in Section 3.5, which show that batch size (gradient noise) affects the optimal value of $p$, we study our example under varying signal-to-noise regimes.

**Slow dynamics along flat directions.** As discussed in Section A, recent studies have revealed key properties of the landscape and training dynamics in LLM training. In particular, the landscape can be decomposed into flat and sharp directions (also referred to as river and valley components (Wen et al., 2025)). The loss along river component typically determines the loss at the bottom of the landscape. Along these flat directions, the optimizer tends to make slow but steady progress, and appears to remain aligned for a period of time.

Motivated by this picture, we consider a one-dimensional example to study whether varying $p$ in AdamPower can *accelerate these slow dynamics along the flat directions*, thereby leading to more efficient loss descent.

**Example 4.1.** *For simplicity, consider a 1-dimensional flat direction. Let the stochastic gradients at time $t \in \mathbb{N}$ follow $g_t \overset{\text{i.i.d.}}{\sim} \text{Unif}(\mu - \sigma, \mu + \sigma)$, where $0 < \mu, \sigma \ll 1$[3]. Here, $\mu$ reflects the full-batch gradient, and $\sigma$ captures the stochastic noise level.*

Our goal is to investigate the values of $p$ that maximize the update magnitude $u_t = m_t/(\sqrt{v_t} + \epsilon)$ in AdamPower (Alg. 1). For simplicity, we set weight decay to 0. We now present both empirical and theoretical analysis.

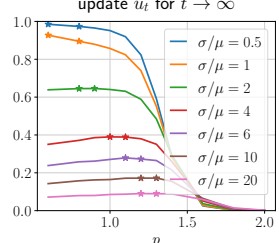

**Empirical findings.** We begin by numerically simulating the update $u_t$. The results are presented in Figure 7. Notably, the optimal value of $p$ varies across noise-to-signal regimes, exhibiting two distinct behaviors:

- *Low-noise regime $\sigma/\mu \leqslant 1$* (blue and orange curves), it is clear that the update magnitude decreases monotonically with increasing $p$, and the optimal power is small, satisfying $p^\star < 1$.
- *High-noise regime $\sigma/\mu > 1$*, the update magnitude increases and then decreases with increasing $p$. Moreover, for noise-dominant regime, the optimal power satisfies $p^\star > 1$ (red, purple, brown, and pink curves).

Figure 7: Numerical results for Example 4.1. We plot the value of $u_t$ at $t = 10^6$ for AdamPower across different $p$'s under varying noise-to-signal ratios. For each curve, the optimal and suboptimal $p$ values are marked with stars. The $\mu$ is set to $\mu = 10^{-6}$. Other hyperparameters follow standard values: $\beta_1 = 0.9$, $\beta_2 = 0.95$, $\epsilon = 10^{-8}$, and $\lambda = 0$. The learning rate $\eta$ does not affect the result.

These findings closely align with our empirical results in real-world LLM pre-training tasks in Section 3. As the batch size increases (corresponding to lower gradient noise), the optimal power $p^\star$ decreases accordingly, transitioning from $p^\star > 1$ to $p^\star < 1$, as observed in Section 3.5. Remarkably, the optimal power $p^\star = 1.2$ in the high-noise regime matches the value used across most LLM pre-training experiments in Section 3.

**Theoretical analysis.** To better understand these interesting behaviors, we theoretically analyze this problem. To facilitate analytical derivation, we consider the limiting case where $\beta_2 \to 1$, which closely approximates typical settings in practice (e.g., 0.95 or 0.999). We define the limiting update of AdamPower as:

$$u := \lim_{t \to \infty} \lim_{\beta_2 \to 1} u_t, \tag{3}$$

where $u_t = m_t/(\sqrt{v_t} + \epsilon)$, with $m_t = \text{EMA}_{\beta_1}(\{g_s^p\}_1^t)$ and $v_t = \text{EMA}_{\beta_2}(\{(g_s^p)^2\}_1^t)$. In this limit, we obtain the *closed-form expression*: $u = \mathbb{E}[g^p]/(\sqrt{\mathbb{E}[(g^p)^2]} + \epsilon)$, *a.s.*, $g \sim \text{Unif}(\mu - \sigma, \mu + \sigma)$. This formulation allows explicit computation and facilitates verification of the empirical trends. We present two propositions corresponding to the low-noise and high-noise regimes.

---

[3]Empirical studies suggest that gradient scales in LLM training are often very small (Huang et al., 2025).

**Proposition 4.2** (low-noise regime, $\sigma \ll \mu$). *It holds that $u = \frac{1+o(1)}{1+\frac{\epsilon}{\mu^p}}$, a.s.. Letting $\tilde{u} = \frac{1}{1+\frac{\epsilon}{\mu^p}}$, we observe that $\tilde{u}$ is monotonically decreasing w.r.t. $p$.*

This proposition quantitatively explains the monotonicity observed in the low-noise regime. Furthermore, it shows that the maximum update is approximately $\frac{1}{1+\epsilon} \approx 1$, achieved in the limit as $p \to 0$. This aligns with Figure 7.

**Proposition 4.3** (high-noise regime, $\mu \ll \sigma$). *It holds that $u = \frac{\mu}{\sigma} \frac{1+o(1)}{\frac{1}{\sqrt{2p+1}} + \frac{\epsilon}{\sigma^p}}$, a.s.. Letting $\tilde{u} = \frac{\mu}{\sigma} \frac{1}{\frac{1}{\sqrt{2p+1}} + \frac{\epsilon}{\sigma^p}}$, we observe the following: If $\epsilon \log(1/\sigma) < 1$, then there exists an optimal power $p^\star$ such that $\tilde{u}$ increases for $0 < p < p^\star$ and decreases for $p > p^\star$. Moreover, we have a tight estimate: $p^\star = \Theta\left(\frac{\log(\epsilon \log(1/\sigma))}{\log \sigma}\right).$*

Notably, in practice, $\epsilon$ is typically chosen sufficiently small (e.g., $\epsilon \ll \sigma$), ensuring $\frac{\log(\epsilon \log(1/\sigma))}{\log \sigma} > 1$. This again aligns with our empirical observation that $p^\star > 1$ in the high-noise regime.

The intuition behind Proposition 4.3 is as follows. When $p$ is relatively small, the denominator is dominated by $\sqrt{\mathbb{E}[(g^p)^2]}$. Since $g \ll 1$, increasing $p$ reduces both the numerator $\mathbb{E}[g^p]$ and denominator $\sqrt{\mathbb{E}[(g^p)^2]}$. In the high-noise regime, the reduction in the denominator outweighs that in the numerator, resulting in a larger update. In contrast, when $p$ is relatively large, the denominator is dominated by $\epsilon$, and AdamPower degenerates to SGDPower, where the update is approximately $\mathbb{E}[g^p]/\epsilon$. In this regime, increasing $p$ reduces the update magnitude.

Although the above example is synthetic, it reveals several non-trivial phenomena highly aligned with LLM pre-training tasks, particularly the existence and behavior of the best $p^\star$ across noise-to-signal regimes. These insights deepen our understanding of how GradPower influences the performance of AdamPower and suggest practical guidance for selecting $p$.

### 4.2 Convergence Guarantees

Let $\mathcal{L} : \mathbb{R}^d \to \mathbb{R}$ be a non-convex loss function. For any $\boldsymbol{\theta} \in \mathbb{R}^d$, let $\boldsymbol{g}(\boldsymbol{\theta})$ denote the stochastic gradient satisfying $\mathbb{E}[\boldsymbol{g}(\boldsymbol{\theta})] = \nabla\mathcal{L}(\boldsymbol{\theta})$.

In this subsection, we consider the classical setting of smooth non-convex optimization and investigate the theoretical benefits of applying GradPower within adaptive optimizers. Since the analysis of Adam is technically complex, to gain clear theoretical insights, we instead analyze its predecessor, Adagrad, a foundational adaptive optimization algorithm (Duchi et al., 2011). The update rule of **AdagradPower** (Adagrad using GradPower) is given by:

$$\boldsymbol{\theta}_{t+1} = \boldsymbol{\theta}_t - \eta \frac{\boldsymbol{g}_t^p}{\sqrt{\boldsymbol{v}_t + \epsilon}}, \quad \boldsymbol{v}_t = \sum_{s=1}^{t} (\boldsymbol{g}_t^p)^2, \tag{4}$$

where the power $p > 0$, and we $\boldsymbol{g}_t$ denotes the stochastic gradient $\boldsymbol{g}(\boldsymbol{\theta}_t)$ for simplicity.

To establish the convergence results, we adopt the following standard assumptions, consistent with Section 2.3 in Défossez et al. (2022).

**Assumption 4.4** (Défossez et al. (2022)). The following conditions hold:

- $\mathcal{L}$ is bounded below by $\mathcal{L}^\star$, i.e., for all $\boldsymbol{\theta} \in \mathbb{R}^d$, $\mathcal{L}(\boldsymbol{\theta}) \geqslant \mathcal{L}^\star$.

- *The loss function is $H$-smooth*, i.e., there exists a constant $H > 0$ such that for all $\boldsymbol{\theta}, \boldsymbol{\theta}' \in \mathbb{R}^d$, $\|\nabla\mathcal{L}(\boldsymbol{\theta}) - \nabla\mathcal{L}(\boldsymbol{\theta}')\|_2 \leqslant H\|\boldsymbol{\theta} - \boldsymbol{\theta}'\|_2$.

- *The $\ell_\infty$ norm of the stochastic gradients is uniformly almost surely bounded*, i.e., there exists a constant $R > 0$ such that for all $\boldsymbol{\theta} \in \mathbb{R}^d$, $\|\boldsymbol{g}(\boldsymbol{\theta})\|_\infty + \epsilon \leqslant R$, a.s..

Under this assumption, the convergence guarantee of Adagrad is well established:

**Theorem 4.5** (Adagrad; Theorem 1 in Défossez et al. (2022)). *Suppose Assumption 4.4 holds. Let $\{\boldsymbol{\theta}_t\}_{t=0}^T$ are trained by **Adagrad** (4) with $p = 1$. Then for any $T \in \mathbb{N}$, we have:*

$$\min_{1 \leqslant t \leqslant T} \mathbb{E}\left[\|\nabla\mathcal{L}(\boldsymbol{\theta}_t)\|_2^2\right] \leqslant \frac{2R(\mathcal{L}(\boldsymbol{\theta}_0) - \mathcal{L}^\star)}{\eta\sqrt{T}} + \frac{Rd(4R + \eta H)\log(1 + R^2 T/\epsilon)}{\sqrt{T}}. \tag{5}$$

We now study the convergence of AdagradPower in both low-noise and high-noise regimes.

**Low-noise regime.** We introduce an additional assumption about the noise scale.

**Assumption 4.6** (Low-noise regime). There exist constants $p \in (0,1)$ and $c > 0$ such that $\mathbb{E}[g_i^p(\boldsymbol{\theta})]\nabla_i\mathcal{L}(\boldsymbol{\theta}) \geqslant c|\nabla_i\mathcal{L}(\boldsymbol{\theta})|^{p+1}$ holds for for all $\boldsymbol{\theta} \in \mathbb{R}^d$ and $i \in [d]$.

This assumption is satisfied in many low-noise scenarios:

**Example 4.7.** *(I) Deterministic regime (the limit case of low noise): if $g_i(\boldsymbol{\theta}) = \nabla_i\mathcal{L}(\boldsymbol{\theta})$, then Assumption 4.6 holds for all $p \in (0,1)$ with $c = 1$. (II) Uniform distribution: if $g_i \sim \mathrm{Unif}(\nabla_i\mathcal{L} - \sigma, \nabla_i\mathcal{L} + \sigma)$ with $\sigma \ll |\nabla_i\mathcal{L}|$, then Assumption 4.6 holds for all $p \in (0,1)$ as $\mathbb{E}[g_i^p]\nabla_i\mathcal{L} = |\nabla_i\mathcal{L}|^{p+1}(1 + o(\sigma/|\nabla_i\mathcal{L}|)) \geqslant 0.99|\nabla_i\mathcal{L}|^{p+1}$.*

**Theorem 4.8** (AdagradPower, low-noise regime). *Suppose Assumption 4.4 and 4.6 hold, as well as $R < 1$[4]. Let $\{\boldsymbol{\theta}_t\}_{t=0}^T$ are trained by **AdagradPower** (4), with the power $p \in (0,1)$ as given in Assumption 4.6. Then for any $T \in \mathbb{N}$, we have:*

$$\min_{1 \leqslant t \leqslant T} \mathbb{E}\left[\|\nabla\mathcal{L}(\boldsymbol{\theta}_t)\|_2^2\right] \leqslant \mathcal{O}\left(\frac{\log^{2/(p+1)} T}{T^{1/(p+1)}}\right). \tag{6}$$

Comparing Theorems 4.5 and 4.8, we observe that AdagradPower achieves a convergence rate $(\log^{2/(p+1)} T/T^{1/(p+1)})$ that is $2/(p+1)$ times faster than Adagrad $(\log T/\sqrt{T})$ in low-noise regime. For Example 4.7, this yields nearly a $2\times$ acceleration for $p \to 0$. This result is consistent with observations in Section 4.1 and 3.5 that the optimal power $p$ for adaptive optimizers is less than 1 in the low-noise regime. The proof is presented in Appendix D.

**High-noise regime.** We introduce an additional assumption regarding the noise scale:

**Assumption 4.9** (High-noise regime). (C1) There exist constants $p > 1, \sigma > 0$ such that $\mathbb{E}[g_i^p(\boldsymbol{\theta})]\nabla_i\mathcal{L}(\boldsymbol{\theta}) \geqslant \sigma|\nabla_i\mathcal{L}(\boldsymbol{\theta})|^2$ holds for all $\boldsymbol{\theta} \in \mathbb{R}^d$ and $i \in [d]$. (C2). It holds that $\sigma > R^{p-1}$.

The first condition asserts that the gradient noise is non-degenerate. The second condition further asserts that the gradient noise is in a high level. Noteworthily, These conditions are naturally satisfied in many high-noise settings:

**Example 4.10.** *Consider $g_i$ satisfy binary distribution $\mathbb{P}(g_i = \nabla_i\mathcal{L} - \sigma_i) = \mathbb{P}(g_i = \nabla_i\mathcal{L} + \sigma_i) = \frac{1}{2}$. Then for any odd number $p > 1$, $\mathbb{E}[g_i^p]\nabla_i\mathcal{L} \geqslant p\sigma_i^{p-1}|\nabla_i\mathcal{L}|^2$. Thus, (C1) in Assumption 4.9 holds with $\sigma = p\sigma_i^{p-1}$. As for (C2), in high-noise regime with $|\nabla_i\mathcal{L}| \ll \sigma_i$, we have $\frac{R^{p-1}}{\sigma} \leqslant \frac{(|\nabla_i\mathcal{L}|+\sigma_i)^{p-1}}{p\sigma_i^{p-1}} \leqslant \frac{1.01}{p} < 1$.*

**Theorem 4.11** (AdagradPower, high-noise regime). *Suppose Assumption 4.4 and 4.9 hold, as well as $R < 1$. Let $\{\boldsymbol{\theta}_t\}_{t=0}^T$ be trained by **AdagradPower** (4), with the power $p > 1$ as given in Assumption 4.9. Then for any $T \in \mathbb{N}$, we have:*

$$\min_{1 \leqslant t \leqslant T} \mathbb{E}\left[\|\nabla\mathcal{L}(\boldsymbol{\theta}_t)\|_2^2\right] \leqslant \frac{R^{p-1}}{\sigma} \cdot \left(\text{R.H.S. of (5)}\right), \tag{7}$$

*where $R^{p-1}/\sigma < 1$.*

Comparing Theorems 4.5 and 4.11, we observe that AdagradPower accelerates convergence of Adagrad by a constant factor $\frac{R^p}{\sigma}$ in high-noise regime. For Example 4.10, the acceleration is significant, due to $\frac{R^{p-1}}{\sigma} \leqslant \frac{1.01}{p}$ for any positive odd $p$. This result provides theoretical support for the empirical superiority of adaptive optimizers using GradPower in LLM pretraining in Section 3 Notably, the theoretical insights are highly aligned with those in Proposition 4.3. In the high-noise regime, using $p > 1$ reduces both numerator $\boldsymbol{g}_t$ and denominator $\sqrt{\boldsymbol{v}_t + \epsilon}$. However, reduction in the denominator outweighs that in the numerator, resulting in a faster convergence speed. The formal proof refines this argument and is presented in Appendix D.

## 5  CONCLUSION

We propose GradPower, a simple yet effective method for improving the efficiency of gradient-based optimizers. Experimentally, AdamPower (Adam using GradPower) consistently achieves lower terminal loss and improved scaling laws than Adam across various LLM pre-training tasks.

---

[4]Empirical studies suggest that gradient scales in LLM training are often very small (Huang et al., 2025).

For future work, it would be interesting to investigate why AdamPower exhibits particular potential for MoE models and `wsd` LR scheduler. Experimentally, exploring the applicability of GradPower beyond LLMs, as well as its integration with other optimizers, could further extend its impact. In addition, developing a dynamic schedule for the GradPower exponent $p$, adapted to the evolving SNR throughout training, presents both a challenging and a potentially valuable direction.

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

# Appendix

## A  RELATED WORKS

**Optimizer disign in LLM pre-training.** In LLM pre-training, Adam (Kingma & Ba, 2014) has become the de facto optimizer. Recent efforts to improve its efficiency focus on two aspects: accelerating convergence and reducing memory usage. Techniques for *accelerating convergence* include introducing curvature information (Liu et al., 2024b; Wang et al., 2024; 2025), mixing momentum (Xie et al., 2024; Pagliardini et al., 2025), variance reduction (Yuan et al., 2024), cautious update (Liang et al., 2024), and applying matrix-based preconditioners (Keller et al., 2024; Vyas et al., 2024). *Memory-efficient* techniques include reducing the moments usage in Adam (Zhang et al., 2025), sign-based updates (Chen et al., 2024b; Liu et al., 2025b), low precision optimizer states (Dettmers et al., 2022; Li et al., 2023), low-rank approximation (Hu et al., 2022; Zhao et al., 2024; Chen et al., 2024a), and structured learning rates (Zhu et al., 2025). Among these, Muon (Keller et al., 2024) stands out for improving both convergence and memory usage and showing strong scalability (Liu et al., 2025a). **In contrast**, our method, GradPower, improves training efficiency without altering the base optimizer's internal updates. Notably, GradPower is orthogonal and complementary to the methods above: it can serve as a lightweight plug-in that further enhances existing optimizers.

**The fast-slow dynamics** in neural network training. Recent works (Wu et al., 2018; Jastrzebski et al., 2020; Cohen et al., 2020; 2022) show that neural network training typically occurs at the so-called Edge of Stability (EoS) stage. This regime is characterized by the optimizer exhibiting

oscillatory behavior along sharp directions without divergence, while steadily progressing along *flat directions*, leading to loss reduction. Several studies (Wen et al., 2025; Song et al., 2025; Cohen et al., 2025; Wang et al., 2024) have emphasized the importance of the slow dynamics along flat directions (referred to as stable direction in Wang et al. (2024), river directions by Wen et al. (2025) and bulk directions by Song et al. (2025)), in *reducing total loss*. Moreover, Wen et al. (2025) further showed that this picture is crucial for understanding the behavior of LLM pre-training. In addition, Fig.3 in (Wen et al., 2025) and Fig.8 (Song et al., 2025) suggest that, the optimizer's trajectory within flat directions tends to *remain aligned* for a period of time.

**Powerball method.** After completing this work, we found that the Powerball method (Yuan et al., 2019) shares the similar methodology as our approach. However, prior studies on Powerball method have been restricted to traditional optimizers—such as GD (Yuan et al., 2019), SGD (Zhou et al., 2020; Yang, 2024), and SARAH (Qin et al., 2025)—and evaluated primarily on relatively small-scale benchmarks including CIFAR-10, CIFAR-100 and MNIST. Although Baiesi (2019) combined Powerball with Adam, the experiments were limited to small and illustrative problems. In contract, our work focuses on modern adaptive optimizers such as Adam and Muon in the context of language model pre-training, a modern and practically important setting. Moreever, previous Powerball studies examined only the narrow regime with $p < 1$, our work studies both $p < 1$ and $p > 1$ regimes, and further develop a comprehensive theoretical study of the relationship between optimal $p$ and batch size.

**Explain the terminology of flat directions.** In classical optimization theory, flat directions refer to Hessian eigenvectors associated with small eigenvalues. However, our usage of flat direction is approximate and follows a line of prior work showing that the **anisotropy of gradient noise** closely reflects the Hessian's curvature structure. Works (Zhu et al., 2019; Wu et al., 2020; Mori et al., 2022; Wu et al., 2022) establish that **directions with small Hessian curvature exhibit low gradient-noise variance, while directions with large Hessian curvature exhibit high gradient-noise variance.** Consequently, flat directions approximately correspond to low-noise directions, and sharp directions to high-noise directions.

# B EXPERIMENTAL DETAILS

**Models.** We utilize two popular classes of LLM models for our pre-training experiments:

- **LLaMA.** LLaMA (Touvron et al., 2023) is a popular Dense decoder-only Transformer architecture, incorporating Rotary Positional Encoding (RoPE) (Su et al., 2024), Swish-Gated Linear Unit (SwiGLU), and Root mean square layer normalization (RMSNorm). We pre-train LLaMA models of sizes ranging from 66M to 2B parameters. Additional model configurations are detailed in Table 3.

- **Qwen2MoE.** Qwen2MoE (Yang et al., 2024a) is a popular open-source MoE decoder-only Transformer architecture. Comparing with Llama, Qwen2MoE utilizes a mix of sliding window and full attention, as well as mixture-of-experts architecture. We disable sliding window attention due to relatively small context length in our experiment. We activate 4 experts per token for all models. For detailed model configurations, refer to Table 4.

**Datasets.** Models are pre-trained on the following datasets:

- **Colossal Clean Crawled Corpus (C4)** (Raffel et al., 2020). It is a large-scale public language dataset, widely used for LLM pre-training such as T5 (Raffel et al., 2020), and prior pre-training studies (Zhao et al., 2024; 2025). We use the T5 tokenizer, with the vocabulary size 32100.

- **OpenWebText** (Gokaslan & Cohen, 2019). It is an opensource recreation of the WebText corpus, is extensively utilized for LLM pre-training such as RoBERTa (Liu et al., 2019) and nanoGPT (Karpathy, 2022). Following Karpathy (2022); Liu et al. (2024b), we use the GPT-2 tokenizer, with the vocabulary size 50304.

**LR schedulers.** We evaluate two popular LR scheduling strategies:

- `cos` (cosine scheduler) (Karpathy, 2022; Touvron et al., 2023): a linear warm-up to peak `lr_max`, followed by cosine decay to a terminal LR `lr_min`.

- `wsd` (warmup-stable-decay scheduler) (Zhai et al., 2022; Hu et al., 2024; Hägele et al., 2024): a linear warm-up LR to peak `lr_max`, followed by a stable phase where LR remains at `lr_max` (up to 80% of the total training steps), and then a linear decay to `lr_min`.

All experiments are conducted on 8 A100 80G GPUs.

Table 3: Dense model configurations and optimally-tuned peak learning rates for Adam.

| Acronym | Size | $d_{\mathrm{model}}$ | $d_{\mathrm{FF}}$ | n_head | depth | lr_max |
|---|---|---|---|---|---|---|
| LLaMA (66M) | 66M | 512 | 2048 | 8 | 8 | 1e-3 (on C4) |
| LLaMA (0.2B) | 200M | 1024 | 4096 | 16 | 8 | 1e-3 (on C4) |
| LLaMA (0.25B) | 237M | 1024 | 4096 | 16 | 8 | 8e-4 (on OpenWebText) |
| LLaMA (0.4B) | 400M | 1280 | 5120 | 16 | 12 | 6e-4 (on C4) |
| LLaMA (1B) | 1004M | 1600 | 6400 | 25 | 22 | 3e-4 (on C4) |
| LLaMA (2B) | 1994M | 2048 | 8096 | 32 | 28 | 2e-4 (on C4) |

Table 4: MoE model configurations and optimally-tuned peak learning rates for Adam on C4.

| Acronym | Size | Activated Size | $d_{\mathrm{model}}$ | $d_{\mathrm{FF}}$ | n_head | depth | n_experts | lr_max |
|---|---|---|---|---|---|---|---|---|
| Qwen2MoE (0.5B) | 502M | 247M | 768 | 3072 | 12 | 12 | 16 | 6e-4 |
| Qwen2MoE (1B) | 1040M | 297M | 768 | 3072 | 12 | 15 | 32 | 3e-4 |
| Qwen2MoE (2B) | 1945M | 536M | 1024 | 4096 | 16 | 16 | 32 | 2e-4 |

For the vision experiment, we used the standard 34 layer ResNet model (He et al., 2016) on the CIFAR-10 dataset (Krizhevsky & Hinton, 2009). We use Adam optimizer and the commonly used `cos` learning rate scheduler.

## B.1 EXPERIMENTAL DETAILS FOR SECTION 3.2 AND 3.3

**Adam baselines.** We use the standard Adam optimizer (with decoupled weight decay) as the baseline in most experiments (expect Section 3.4). The baseline is configured with hyperparameters $\beta_1 = 0.9, \beta_2 = 0.95$, weight decay $\lambda = 0.1$, and gradient clipping threshold of 1.0, following protocols used in LLaMA pre-training (Touvron et al., 2023). Following Hoffmann et al. (2022), the final learning rate `lr_min` is set to $1/10$ of the peak learning rate `lr_max`. Additionally,

- **C4 pre-training.** We follow the setup of Zhao et al. (2024); Chen et al. (2024a); Zhu et al. (2025), using a sequence length of 256 and batch size of 512. Following the Chinchilla scaling law (Hoffmann et al., 2022), the total number of training tokens is set to be approximately 20 times the number of model parameters. The training includes 1,000 warm-up steps. The grid search for `lr_max` is performed over {1e-4, 2e-4, 3e-4, 6e-4, 1e-3, 1.5e-3}. Optimal learning rates for each model are detailed in Tables 3 and 4.

- **OpenWebText pre-training.** The (max) sequence length is set to 1024, and the batch size is set to 480, following nanoGPT (Karpathy, 2022) and Liu et al. (2024b). The total training duration is 50,000 or 100,000 steps, including 1,000 warm-up steps. The grid search for `lr_max` is performed over {2e-4, 4e-4, 6e-4, 8e-4, 1e-3}. Optimal learning rates for each model are detailed in Table 3.

**AdamPower experiments.** We adopt $p = 1.2$ as the default in all experiments in Section 3.2 and 3.3. All other optimizer hyperparameters are kept identical to those used for the Adam baselines. Importantly, the power $p = 1.2$ proves to be **highly robust**.

## B.2 EXPERIMENTAL DETAILS FOR SECTION 3.4

**Adam with Blockwise LR.** Following Wang et al. (2025), we adopt the same peak `lr_max` tuned for Adam as the `lr_max` of Adam with Blockwise LR. For the blockwise lr ratios, we adopt the recommended $r(\mathrm{Embed}) = 10, r(\mathrm{QK}) = 8, r(\mathrm{FFN}) = 6, r(\mathrm{VO}) = 4$ in Wang et al. (2025).

**AdamPower with Blockwise LR.** We still adopt $p = 1.2$ in the AdamPower with Blockwise LR. All other optimizer hyperparameters are kept identical to those used for the Adam with Blockwise LR.

**Muon baseline.** We use the same techniques for Muon as Liu et al. (2025a): (1) adding weight decay (2) adjusting the per-parameter update scale. These techniques allow our Muon experiment to use the identical learning rate as the Adam baseline without the extra effort of hyper-parameter tuning.

**MuonPower.** We still adopt $p = 1.2$ in the MuonPower. All other optimizer hyperparameters are kept identical to those used for the Muon baseline.

### B.3 EXPERIMENTAL DETAILS FOR SECTION 3.5

We conduct experiments using LLaMA (0.2B) on C4 dataset with `wsd` scheduler. Unlike the previous experimental settings, here we vary the batch size from the standard 512 up to 8192.

For batch size 512, the tuned `max_lr` is `1e-3` (Table 3). For larger batch sizes (2048, 4096, 8192), we tune the `max_lr` over $\{$`6r-4, 1e-3, 2e-3, 4e-3, 8e-3`$\}$ for Adam. We find that `1e-3` consistently yields the best results across all batch sizes.

For each batch size, we evaluate AdamPower with multiple values of $p$, and record their validation loss when the optimal validation loss reaches approximately 3.5.

We also conduct vision experiments using ResNet-34 on CIFAR datset with `cos` scheduler. We tune the `max_lr` over $\{$`6.25e-5, 1.25e-4, 2.5e-4, 5e-4, 1e-3`$\}$. For batch size 32, 128, and 512, the tuned `max_lr` is `1.25e-4, 2.5e-4, 5e-4`, respectively.

### B.4 ADDITIONAL EXPERIMENTS WITH MULTIPLE RANDOM SEEDS

In this subsection, we reproduce a subset of experiments in Figure 3 with multiple random seeds to assess statistical robustness. Specifically, we rerun the experiments six times with different random seeds and report both mean and standard deviation as shown in Figure 8. The shaded regions in the plots denote the standard deviation, showing the statistical significance of each method. These results confirm that the observed performance differences are consistent and cannot be explained by random seed variability.

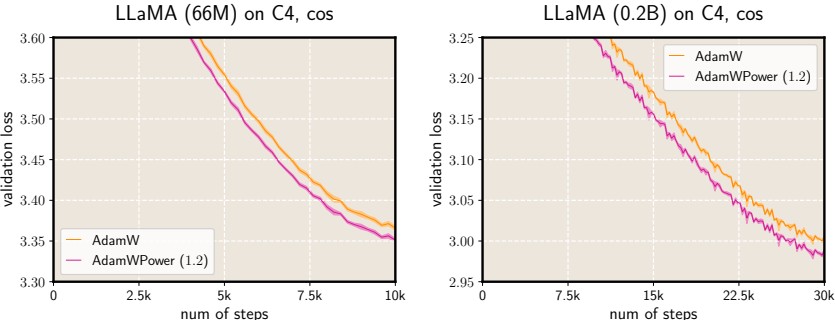

Figure 8: AdamPower ($p = 1.2$) consistently outperforms Adam in LLaMA pre-training tasks. The shaded regions in the plots denote the standard deviation.

### B.5 ADDITIONAL EXPERIMENTS WITH A FINER-GRAINED LEARNING RATE SWEEP

In this subsection, we reproduce a subset of experiments in Figure 1 with a finer-grained learning rate sweep. Specifically, we use $0.94 \times$ the baseline maximum learning rate in "AdamW (0.94lr)" to isolate the contribution of GradPower from these two protential effects:

- **global damping.** As $|g| < 1$, $|g|^p (p > 1)$ induces additional damping of the gradient.

- **heavier tails.** $|g|^p (p > 1)$ suppresses gradients of small magnitude more aggressively than large ones.

The 0.94 factor approximates the expected update magnitude ratio between Adam and AdamPower with $p = 1.2$ under zero-mean Gaussian gradients. As shown in Figure 9 and Table 5, AdamPower with $p = 1.2$ continuously surpasses Adam within this finer-grained learning rate sweep.

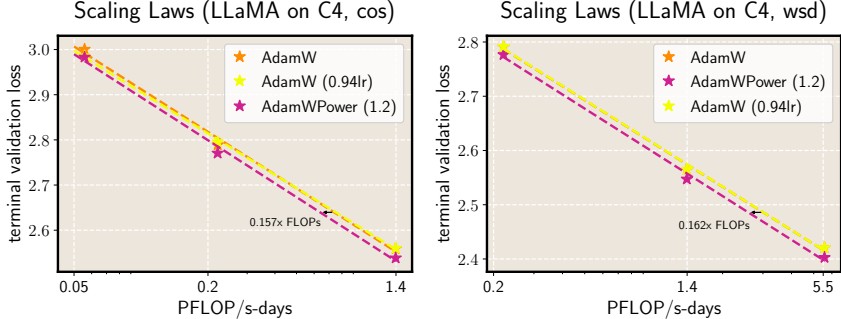

Figure 9: Scaling-law comparison of AdamPower and Adam on the C4 dataset for dense LLaMA models within a finer-grained learning rate sweep.

| setting | LLaMA on C4, cos | | | LLaMA on C4, wsd | | |
|---|---|---|---|---|---|---|
| model size | 0.2B | 0.4B | 1B | 0.4B | 1B | 2B |
| AdamW | 3.0006 | 2.7889 | 2.5593 | 2.7911 | 2.5645 | 2.4206 |
| AdamW (0.94lr) | 2.9859 | 2.7957 | 2.5601 | 2.7917 | 2.5649 | 2.4207 |
| AdamWPower (1.2) | **2.9832** | **2.7705** | **2.5385** | **2.7767** | **2.5472** | **2.4028** |

Table 5: Scaling-law comparison of AdamPower and Adam on the C4 dataset for dense LLaMA models within a finer-grained learning rate sweep.

## B.6 INTERACTION BETWEEN GRADPOWER AND GRADIENT CLIPPING

In this subsection, we examine the ordering of gradient clipping and the GradPower transformation. Gradient clipping is a standard component in LLM pre-training pipelines, and in our default setup, gradient clipping is applied first, followed by the GradPower transformation. Notably, both orderings yield bounded gradients, ensuring that the two procedures remain comparable from a stability standpoint.

To directly evaluate the interaction, we conduct a controlled experiment based on the setting of Figure 8 on LLaMA-0.2B (dense). In a controlled manner, we switch the order of gradient clipping and the GradPower transformation. We refer to this variant as AdamWPower-II, in contrast to the standard AdamWPower implementation. As shown in Figure 10, the training curves are nearly indistinguishable across the full training trajectory, indicating that the ordering does not materially affect performance.

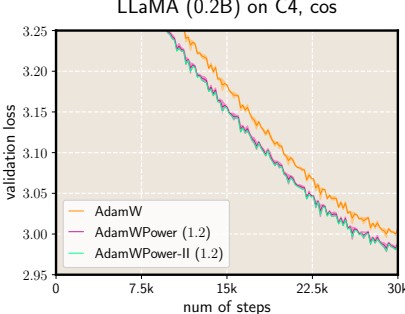

Figure 10: AdamPower ($p = 1.2$) outperforms Adam in LLaMA pre-training tasks. The shaded regions in the plots denote the standard deviation.

## C  PROOFS IN SECTION 2 AND 4.1

### C.1  SUPPORT FOR THE MOTIVATION IN SECTION 2

In this section, we provide a detailed justification for the claim in Section 2 that the **linear transformation** ($\varphi(z) = cz$ with $c \in \mathbb{R}$) **fails** to alter the updates of popular optimizers used in LLM pretraining. Without loss of generality, we analyze the one-dimensional case.

- **Adaptive optimizers**, including Adagrad, RMSprop, and Adam. These optimizers adjust the learning rate based on a moving average of gradients. In practice, the term $\epsilon$ (used to ensure numerical stability) is typically set to an extremely small value (e.g., `1e-8, 1e-12`), Consider the update rule of Adam in the limit $\epsilon \to 0$:

$$\theta_{t+1} = (1 - \lambda \eta_t)\theta_t - \eta_t \frac{\text{EMA}_{\beta_1}(\{g_s\}_1^t)}{\sqrt{\text{EMA}_{\beta_2}(\{g_s^2\}_1^t)}}.$$

  Applying a linear transformation $\varphi(z) = cz$ with $c > 0$ does not change the ratio:

$$\frac{\text{EMA}_{\beta_1}(\{g_s\}_1^t)}{\sqrt{\text{EMA}_{\beta_2}(\{g_s^2\}_1^t)}} = \frac{\text{EMA}_{\beta_1}(\{\varphi(g_s)\}_1^t)}{\sqrt{\text{EMA}_{\beta_2}(\{\varphi(g_s)^2\}_1^t)}}.$$

  Hence, the dynamics remains unchanged. This argument applies similarly to Adagrad and RMSprop.

- **Sign-based methods**, including Sign momentum (Bernstein et al., 2018) and Lion (Chen et al., 2024b). These methods operate on the sign of the moving average gradients. For instance, Signed Momentum (with decoupled weight decay) follows:

$$\theta_{t+1} = (1 - \lambda \eta_t)\theta_t - \eta_t \text{sign}(\text{EMA}_\beta(\{g_s\}_1^t)).$$

  Again, applying a linear transformation $\varphi(z) = cz$ with $c > 0$ does not change the sign of the averaged gradient, since:

$$\text{sign}(\text{EMA}_\beta(\{g_s\}_1^t)) = \text{sign}(\text{EMA}_\beta(\{\varphi(g_s)\}_1^t)).$$

  Hence, the dynamics remains unchanged. This argument applies similarly to Lion.

In contrast, our proposed (nonlinear) GradPower transformation ($\varphi_p(z) = z^p := |z|^p \text{sign}(z)$ with $p > 0$) *does* alter the updates of both adaptive and sign-based optimizers, when the gradients $g_s$ are not all of the same sign.

### C.2  PROOF OF PROPOSITIONS 4.2 AND 4.3

$$\mathbb{E}[\varphi_p(g)] = \frac{(\mu + \sigma)^{p+1} - |\mu - \sigma|^{p+1}}{2\sigma(p+1)},$$

$$\mathbb{E}[\varphi_p^2(g)] = \frac{(\mu + \sigma)^{2p+1} - |\mu - \sigma|^{2p+1} \text{sign}(\mu - \sigma)}{2\sigma(2p+1)}.$$

The low-noise regime. ($0 \ll \sigma \ll \mu \ll 1$)

It is straightforward that

$$\mathbb{E}[\varphi_p(g)] = \frac{\mu^{p+1}}{2\sigma(p+1)} \left( \left(1 + \frac{\sigma}{\mu}\right)^{p+1} - \left(1 - \frac{\sigma}{\mu}\right)^{p+1} \right)$$

$$= \frac{\mu^{p+1}}{2\sigma(p+1)} \left( \frac{2(p+1)\sigma}{\mu} + o\left(\frac{\sigma}{\mu}\right) \right) = \mu^p (1 + o(1));$$

$$\mathbb{E}[\varphi_p^2(g)] = \frac{\mu^{2p+1}}{2\sigma(2p+1)} \left( \left(1 + \frac{\sigma}{\mu}\right)^{2p+1} - \left(1 - \frac{\sigma}{\mu}\right)^{2p+1} \right)$$

$$=\frac{\mu^{2p+1}}{2\sigma(2p+1)}\left(\frac{2(2p+1)\sigma}{\mu}+o\left(\frac{\sigma}{\mu}\right)\right)=\mu^{2p}\left(1+o(1)\right).$$

Therefore, we have

$$u=\frac{\mathbb{E}[\varphi_p(g)]}{\sqrt{\mathbb{E}[\varphi_p^2(g)]}+\epsilon}=\frac{\mu^p\left(1+o(1)\right)}{\mu^p\left(1+o(1)\right)+\epsilon}=\frac{1+o(1)}{1+\frac{\epsilon}{\mu^p}}.$$

The high-noise regime. ($0\ll\mu\ll\sigma\ll1$)

It is straightforward that

$$\mathbb{E}[\varphi_p(g)]=\frac{\sigma^{p+1}}{2\sigma(p+1)}\left(\left(1+\frac{\mu}{\sigma}\right)^{p+1}-\left(1-\frac{\mu}{\sigma}\right)^{p+1}\right)$$

$$=\frac{\sigma^p}{2(p+1)}\left(\frac{2(p+1)\mu}{\sigma}+o\left(\frac{\mu}{\sigma}\right)\right)=\sigma^{p-1}\mu(1+o(1));$$

$$\mathbb{E}[\varphi_p^2(g)]=\frac{\sigma^{2p+1}}{2\sigma(2p+1)}\left(\left(1+\frac{\mu}{\sigma}\right)^{2p+1}+\left(1-\frac{\mu}{\sigma}\right)^{2p+1}\right)$$

$$=\frac{\sigma^{2p+1}}{2\sigma(2p+1)}\left(2+o\left(\frac{\mu}{\sigma}\right)\right)=\frac{\sigma^{2p}}{2p+1}(1+o(1)).$$

Therefore, we have

$$u=\frac{\mathbb{E}[\varphi_p(g)]}{\sqrt{\mathbb{E}[\varphi_p^2(g)]}+\epsilon}=\frac{\sigma^{p-1}\mu(1+o(1))}{\frac{\sigma^p}{\sqrt{2p+1}}(1+o(1))+\epsilon}=\frac{\mu}{\sigma}\frac{1+o(1)}{\frac{1+o(1)}{\sqrt{2p+1}}+\frac{\epsilon}{\sigma^p}}=\frac{\mu}{\sigma}\frac{1+o(1)}{\frac{1}{\sqrt{2p+1}}+\frac{\epsilon}{\sigma^p}}.$$

To study the monotonicity of $\tilde{u}=\frac{\mu}{\sigma}\frac{1}{\frac{1}{\sqrt{2p+1}}+\frac{\epsilon}{\sigma^p}}$, we only need to study the monotonicity of

$$\psi(p)=\frac{1}{\sqrt{2p+1}}+\frac{\epsilon}{\sigma^p}.$$

It is clear that

$$\psi'(p)=\frac{\epsilon\log(1/\sigma)}{\sigma^p}-\frac{1}{(2p+1)^{3/2}}.$$

Due to $\sigma\log(1/\sigma)<1$, there exists a $p^\star$, such that $\psi'(p)<0$ for all $0<p<p^\star$; $\psi'(p)>0$ for all $p>p^\star$. Here, $p^\star$ is the solution of the equation:

$$\frac{\sigma^p}{(2p+1)^{3/2}}=\epsilon\log(1/\sigma)$$

Noticing the relationship between $\psi$ and $\tilde{u}$, we have: $\tilde{u}$ increases when $0<p<p^\star$; $\tilde{u}$ decreases when $p>p^\star$.

Now we estimate $p^\star$. Due to $1+x\leqslant e^x$, we have $(2p+1)^{3/2}\leqslant(e^{2p})^{3/2}=(e^3)^p$. Then we obtain the two-sides estimate $1\leqslant(2p+1)^{3/2}\leqslant(e^3)^p$, implying

$$\left(\frac{\sigma}{e^3}\right)^p\leqslant\frac{\sigma^p}{(2p+1)^{3/2}}\leqslant\sigma^p.$$

Therefore, we have the estimate:

$$\frac{\log(\epsilon\log(1/\sigma))}{\log(\sigma/e^3)}\leqslant p^\star\leqslant\frac{\log(\epsilon\log(1/\sigma))}{\log\sigma}$$

Noticing $\sigma\ll1$, we obtain:

$$p^\star=\Theta\left(\frac{\log(\epsilon\log(1/\sigma))}{\log\sigma}\right).$$

## D    PROOFS IN SECTION 4.2

Recall that the udpate rule of AdagradPower (with power $p$) follows:

$$\boldsymbol{\theta}_{t+1} = \boldsymbol{\theta}_t - \eta \boldsymbol{u}_t,$$

$$\boldsymbol{u}_t = \frac{\varphi_p(\boldsymbol{g}_t)}{\sqrt{\boldsymbol{v}_t + \epsilon}},$$

$$\boldsymbol{v}_t = \sum_{s=1}^{t} \varphi_p^2(\boldsymbol{g}_t).$$

In general, our proof is inspired by the main techniques to prove Adagrad used in Défossez et al. (2022). The key difference is to establish a similar estimate of the loss descent for Adamgradpower. This generalize is not trivial, need to use the structure of the high-noise fact.

In the proof, we need an auxiliary sequence, defined as:

$$\tilde{\boldsymbol{v}}_t = \boldsymbol{v}_{t-1} + \mathbb{E}_t[\varphi_p^2(\boldsymbol{g}_t)].$$

### D.1    KEY LEMMAS

We need two important lemmas in the proof of each Theorem. The first develops the lower bound of the descent value for the update.

**Lemma D.1** (Descent estimate for the update, high-noise regime)**.** *Under Assumption 4.4, for all $t \in \mathbb{N}$, and $i \in [d]$ and any $\sigma > 0$, we have:*

$$\mathbb{E}_t\left[\nabla_i\mathcal{L}(\boldsymbol{\theta})u_{t,i}\right] = \mathbb{E}_t\left[\frac{\nabla_i\mathcal{L}(\boldsymbol{\theta})\varphi_p(g_{t,i})}{\sqrt{v_{t,i} + \epsilon}}\right] \geqslant \frac{\mathbb{E}_t\left[\nabla_i\mathcal{L}(\boldsymbol{\theta})\varphi_p(g_{t,i})\right]}{\sqrt{\tilde{v}_{t,i} + \epsilon}} - \frac{\sigma}{2}\frac{|\nabla_i\mathcal{L}(\boldsymbol{\theta})|^2}{\sqrt{\tilde{v}_{t,i} + \epsilon}} - \frac{2R^p}{\sigma}\mathbb{E}\left[\frac{\varphi_p^2(g_{t,i})}{v_{t,i} + \epsilon}\right].$$

*Proof of Lemma D.1.*
Let $t \in \mathbb{N}$ and $i \in [p]$. For simplicity, we use the following notations in the proof:

$$G = \nabla_i\mathcal{L}(\boldsymbol{\theta}),\ g = g_{t,i},\ v = v_{t,i},\ \tilde{v} = v_{t,i}.$$

First, we decouple the descent quantity as:

$$\mathbb{E}_t\left[\frac{G\varphi_p(g)}{\sqrt{v + \epsilon}}\right] = \mathbb{E}_t\left[\frac{G\varphi_p(g)}{\sqrt{\tilde{v} + \epsilon}}\right] + \mathbb{E}_t\left[\underbrace{G\varphi_p(g)\left(\frac{1}{\sqrt{v + \epsilon}} - \frac{1}{\sqrt{\tilde{v} + \epsilon}}\right)}_{I}\right] \tag{8}$$

Then we bound the term $I$ in the RHS of Equation (8):

$$|I| = |G\varphi_p(g)|\frac{|\tilde{v} - v|}{\sqrt{v + \epsilon}\sqrt{\tilde{v} + \epsilon}(\sqrt{v + \epsilon} + \sqrt{\tilde{v} + \epsilon})}$$

$$= |G\varphi_p(g)|\frac{|\mathbb{E}_t[\varphi_p^2(g)] - \varphi_p^2(g)|}{\sqrt{v + \epsilon}\sqrt{\tilde{v} + \epsilon}(\sqrt{v + \epsilon} + \sqrt{\tilde{v} + \epsilon})}$$

$$\leqslant |G\varphi_p(g)|\frac{\mathbb{E}_t[\varphi_p^2(g)] + \varphi_p^2(g)}{\sqrt{v + \epsilon}\sqrt{\tilde{v} + \epsilon}(\sqrt{v + \epsilon} + \sqrt{\tilde{v} + \epsilon})}$$

$$\leqslant \underbrace{|G\varphi_p(g)|\frac{\mathbb{E}_t[\varphi_p^2(g)]}{\sqrt{v + \epsilon}(\tilde{v} + \epsilon)}}_{I_1} + \underbrace{|G\varphi_p(g)|\frac{\varphi_p^2(g)}{(v + \epsilon)\sqrt{\tilde{v} + \epsilon}}}_{I_2}.$$

Consequently, we will estimate $I_1$ and $I_2$ by the inequality

$$|xy| \leqslant \frac{\lambda x^2}{2} + \frac{y^2}{2\lambda}.$$

For $I_1$, by taking

$$|x| = \frac{|G|}{\sqrt{\tilde{v} + \epsilon}}, \ |y| = \frac{|\varphi_p(g)|\mathbb{E}_t[\varphi_p^2(g)]}{\sqrt{v + \epsilon}\sqrt{\tilde{v} + \epsilon}}, \ \lambda = \frac{\sigma\sqrt{\tilde{v} + \epsilon}}{2},$$

we obtain

$$I_1 \leqslant \frac{\sigma}{4}\frac{|G|^2}{\sqrt{\tilde{v} + \epsilon}} + \frac{1}{\sigma}\frac{(\varphi_p^2(g)(\mathbb{E}_t[\varphi_p^2(g)])^2}{(v + \epsilon)(\tilde{v} + \epsilon)^{3/2}},$$

$$\mathbb{E}_t[I_1] \leqslant \frac{\sigma}{4}\frac{|G|^2}{\sqrt{\tilde{v} + \epsilon}} + \frac{1}{\sigma}\frac{(\mathbb{E}_t[\varphi_p^2(g)])^2}{(\tilde{v} + \epsilon)^{3/2}}\mathbb{E}_t\left[\frac{\varphi_p^2(g)}{v + \epsilon}\right].$$

Given that $\sqrt{\mathbb{E}_t[\varphi_p^2(g)]} \leqslant \sqrt{\tilde{v} + \epsilon}$ and $\sqrt{\mathbb{E}_t[\varphi_p^2(g)]} \leqslant R^p$, we can simplify the above estimate as:

$$\mathbb{E}_t[I_1] \leqslant \frac{\sigma}{4}\frac{|G|^2}{\sqrt{\tilde{v} + \epsilon}} + \frac{R^p}{\sigma}\mathbb{E}_t\left[\frac{\varphi_p^2(g)}{v + \epsilon}\right].$$

For $I_2$, by taking

$$|x| = \frac{|G|}{\sqrt{\tilde{v} + \epsilon}}, \ |y| = \frac{|\varphi_p(g)|\varphi_p^2(g)}{v + \epsilon}, \ \lambda = \frac{\sigma\varphi_p^2(g)}{2\mathbb{E}_t[\varphi_p^2(g)]},$$

we obtain

$$I_2 \leqslant \frac{\sigma}{4}\frac{\varphi_p^2(g)}{\mathbb{E}_t[\varphi_p^2(g)]}\frac{|G|^2}{\sqrt{\tilde{v} + \epsilon}} + \frac{1}{\sigma}\frac{\mathbb{E}_t[\varphi_p^2(g)]}{\sqrt{\tilde{v} + \epsilon}}\frac{\varphi_p^4(g)}{(v + \epsilon)^2}$$

Given that $\varphi_p^2(g) \leqslant v + \epsilon$, we can simplify the above estimate as:

$$I_2 \leqslant \frac{\sigma}{4}\frac{\varphi_p^2(g)}{\mathbb{E}_t[\varphi_p^2(g)]}\frac{|G|^2}{\sqrt{\tilde{v} + \epsilon}} + \frac{1}{\sigma}\frac{\mathbb{E}_t[\varphi_p^2(g)]}{\sqrt{\tilde{v} + \epsilon}}\frac{\varphi_p^2(g)}{v + \epsilon}.$$

Using $\sqrt{\mathbb{E}_t[\varphi_p^2(g)]} \leqslant \sqrt{\tilde{v} + \epsilon}$, $\sqrt{\mathbb{E}_t[\varphi_p^2(g)]} \leqslant R^p$, and taking the conditional expectation, we obtain:

$$\mathbb{E}_t[I_2] \leqslant \frac{\sigma}{4}\frac{|G|^2}{\sqrt{\tilde{v} + \epsilon}} + \frac{R^p}{\sigma}\mathbb{E}\left[\frac{\varphi_p^2(g)}{v + \epsilon}\right].$$

Consequently, combing the two estimates of $I_1$ and $I_2$, we obtain:

$$\mathbb{E}_t[|I|] \leqslant \mathbb{E}_t[I_1] + \mathbb{E}_t[I_2] \leqslant \frac{\sigma}{2}\frac{|G|^2}{\sqrt{\tilde{v} + \epsilon}} + \frac{2R^p}{\sigma}\mathbb{E}\left[\frac{\varphi_p^2(g)}{v + \epsilon}\right].$$

Putting the above estimate into Equation (8), we obtain the lower bound:

$$\mathbb{E}_t\left[\frac{G\varphi_p(g)}{\sqrt{v + \epsilon}}\right] = \mathbb{E}_t\left[\frac{G\varphi_p(g)}{\sqrt{\tilde{v} + \epsilon}}\right] + \mathbb{E}_t[I] \geqslant \mathbb{E}_t\left[\frac{G\varphi_p(g)}{\sqrt{\tilde{v} + \epsilon}}\right] - \mathbb{E}_t[|I|]$$

$$\geqslant \frac{\mathbb{E}_t[G\varphi_p(g)]}{\sqrt{\tilde{v} + \epsilon}} - \frac{\sigma}{2}\frac{|G|}{\sqrt{\tilde{v} + \epsilon}} - \frac{2R^p}{\sigma}\mathbb{E}_t\left[\frac{\varphi_p^2(g)}{v + \epsilon}\right].$$

$$\square$$

The second lemma estimate the sum of the updates in adaptive methods.

**Lemma D.2** (Lemma 5.2 in Défossez et al. (2022)). *Let $\{a_t\}_{t\in\mathbb{N}}$ be a non-negative sequence, $\epsilon > 0$. Then for all $T \in \mathbb{N}$, we have:*

$$\sum_{t=1}^{T}\frac{a_t}{\epsilon + \sum_{s=1}^{t}a_s} \leqslant \log\left(1 + \frac{1}{\epsilon}\sum_{t=1}^{T}a_t\right).$$

### D.2 PROOF OF THEOREM 4.8

With the help of the above Lemma D.1 and D.2, we can prove Theorem 4.8.

*Proof of Theorem 4.8.*
Due to the $H$-smoothness, we have the quadratic upper bound:

$$\mathcal{L}(\boldsymbol{\theta}_{t+1}) \leqslant \mathcal{L}(\boldsymbol{\theta}_t) - \eta \langle \nabla \mathcal{L}(\boldsymbol{\theta}_t), \boldsymbol{u}_t \rangle + \frac{\eta^2 H}{2} \|\boldsymbol{u}_t\|_2^2.$$

Taking the expectation at $t$, we have:

$$\mathbb{E}_t \left[ \mathcal{L}(\boldsymbol{\theta}_{t+1}) \right] \leqslant \mathcal{L}(\boldsymbol{\theta}_t) - \eta \mathbb{E}_t \left[ \langle \nabla \mathcal{L}(\boldsymbol{\theta}_t), \boldsymbol{u}_t \rangle \right] + \frac{\eta^2 H}{2} \mathbb{E}_t \left[ \|\boldsymbol{u}_t\|_2^2 \right]$$

$$= \mathcal{L}(\boldsymbol{\theta}_t) - \eta \sum_{i=1}^{d} \mathbb{E}_t \left[ \nabla_i \mathcal{L}(\boldsymbol{\theta}_t) u_{t,i} \right] + \sum_{i=1}^{d} \frac{\eta^2 H}{2} \mathbb{E}_t \left[ u_{t,i}^2 \right].$$

Combine Lemma D.1 with $\sigma = c$ and Assumption 4.6, we get

$$\mathbb{E}_t \left[ \nabla_i \mathcal{L}(\boldsymbol{\theta}) u_{t,i} \right] = \mathbb{E}_t \left[ \frac{\nabla_i \mathcal{L}(\boldsymbol{\theta}) \varphi_p(g_{t,i})}{\sqrt{v_{t,i} + \epsilon}} \right] \geqslant c \frac{|\nabla_i \mathcal{L}(\boldsymbol{\theta})|^{p+1}}{\sqrt{\tilde{v}_{t,i} + \epsilon}} - \frac{c}{2} \frac{|\nabla_i \mathcal{L}(\boldsymbol{\theta})|^2}{\sqrt{\tilde{v}_{t,i} + \epsilon}} - \frac{2R^p}{c} \mathbb{E} \left[ \frac{\varphi_p^2(g_{t,i})}{v_{t,i} + \epsilon} \right]$$

$$\geqslant \frac{c}{2} \frac{|\nabla_i \mathcal{L}(\boldsymbol{\theta})|^{p+1}}{\sqrt{\tilde{v}_{t,i} + \epsilon}} - \frac{2R^p}{c} \mathbb{E} \left[ \frac{\varphi_p^2(g_{t,i})}{v_{t,i} + \epsilon} \right].$$

Where last inequality comes from $R < 1$. Using it for each dimension, we have:

$$\mathbb{E}_t \left[ \mathcal{L}(\boldsymbol{\theta}_{t+1}) \right] \leqslant \mathcal{L}(\boldsymbol{\theta}_t) - \frac{\eta c}{2} \frac{|\nabla_i \mathcal{L}(\boldsymbol{\theta}_t)|^{p+1}}{\sqrt{\tilde{v}_{t,i} + \epsilon}} + \frac{2\eta R^p}{c} \mathbb{E} \left[ \frac{\varphi_p^2(g_{t,i})}{v_{t,i} + \epsilon} \right] + \sum_{i=1}^{d} \frac{\eta^2 H}{2} \mathbb{E}_t \left[ u_{t,i}^2 \right]$$

$$= \mathcal{L}(\boldsymbol{\theta}_t) - \sum_{i=1}^{d} \frac{\eta c}{2} \frac{|\nabla_i \mathcal{L}(\boldsymbol{\theta}_t)|^{p+1}}{\sqrt{\tilde{v}_{t,i} + \epsilon}} + \sum_{i=1}^{d} \left( \frac{2\eta R^p}{c} + \frac{\eta^2 H}{2} \right) \mathbb{E}_t \left[ \frac{\varphi_p^2(g_{t,i})}{v_{t,i} + \epsilon} \right].$$

Noticing $\sqrt{\tilde{v}_{t_i} + \epsilon} \leqslant R^p \sqrt{t}$, we further have:

$$\mathbb{E}_t \left[ \mathcal{L}(\boldsymbol{\theta}_{t+1}) \right] \leqslant \mathcal{L}(\boldsymbol{\theta}_t) - \frac{\eta c}{2} \frac{\|\nabla \mathcal{L}(\boldsymbol{\theta}_t)\|_{p+1}^{p+1}}{R^p \sqrt{t}} + \sum_{i=1}^{d} \left( \frac{2\eta R^p}{c} + \frac{\eta^2 H}{2} \right) \mathbb{E}_t \left[ \frac{\varphi_p^2(g_{t,i})}{v_{t,i} + \epsilon} \right].$$

Summing the previous inequality for all $0 \leqslant t \leqslant T - 1$, taking the complete expectation, and using $\sqrt{t} \leqslant \sqrt{T}$, we have:

$$\mathbb{E} \left[ \mathcal{L}(\boldsymbol{\theta}_t) \right] \leqslant \mathcal{L}(\boldsymbol{\theta}_0) - \frac{\eta c \sum_{t=1}^{T} \|\nabla \mathcal{L}(\boldsymbol{\theta}_t)\|_{p+1}^{p+1}}{2\eta R^p \sqrt{T}} + \sum_{i=1}^{d} \left( \frac{2R^p}{c} + \frac{\eta^2 H}{2} \right) \mathbb{E} \left[ \sum_{t=1}^{T} \frac{\varphi_p^2(g_{t,i})}{v_{t,i} + \epsilon} \right].$$

Then for each dimension, using Lemma D.2 for the sequence $\{(g_{t,i}^p)^2\}_{1 \leqslant t \leqslant T}$, we obtain:

$$\mathbb{E} \left[ \mathcal{L}(\boldsymbol{\theta}_t) \right]$$

$$\leqslant \mathcal{L}(\boldsymbol{\theta}_0) - \frac{\eta c \sum_{t=1}^{T} \mathbb{E} \|\nabla \mathcal{L}(\boldsymbol{\theta}_t)\|_{p+1}^{p+1}}{2R^p \sqrt{T}} + \left( \frac{2\eta R^p}{c} + \frac{\eta^2 H}{2} \right) d \, \mathbb{E} \left[ \log \left( 1 + \frac{1}{\epsilon} \sum_{t=1}^{T} \varphi_p^2(g_{t,i}) \right) \right]$$

$$\leqslant \mathcal{L}(\boldsymbol{\theta}_0) - \frac{\eta c \sum_{t=1}^{T} \mathbb{E} \|\nabla \mathcal{L}(\boldsymbol{\theta}_t)\|_{p+1}^{p+1}}{2R^p \sqrt{T}} + \left( \frac{2\eta R^p}{c} + \frac{\eta^2 H}{2} \right) d \log \left( 1 + \frac{R^{2p}}{\epsilon} T \right).$$

This implies:

$$\mathbb{E} \min_{1 \leqslant t \leqslant T} \|\nabla \mathcal{L}(\boldsymbol{\theta}_t)\|_{p+1}^{p+1} \leqslant \frac{1}{T} \sum_{t=1}^{T} \mathbb{E} \|\nabla \mathcal{L}(\boldsymbol{\theta}_t)\|_{p+1}^{p+1}$$

$$\leqslant \frac{2R^p}{c\sqrt{T}} \left( \frac{\mathcal{L}(\boldsymbol{\theta}_0) - \mathcal{L}^\star}{\eta} + \left( \frac{2R^p}{c} + \frac{\eta H}{2} \right) d \log \left( 1 + \frac{R^{2p}}{\epsilon} T \right) \right) = \mathcal{O} \left( \frac{\log T}{\sqrt{T}} \right).$$

Hence

$$\min_{1 \leqslant t \leqslant T} \|\nabla \mathcal{L}(\boldsymbol{\theta}_t)\|_2^2 \leqslant (\min_{1 \leqslant t \leqslant T} \|\nabla \mathcal{L}(\boldsymbol{\theta}_t)\|_{p+1}^{p+1})^{2/(p+1)} = \mathcal{O} \left( \frac{\log^{2/(p+1)} T}{T^{1/(p+1)}} \right).$$

$\square$

### D.3 PROOF OF THEOREM 4.11

With the help of the above Lemma D.1 and D.2, we can prove Theorem 4.11.

*Proof of Theorem 4.11.*
Due to the $H$-smoothness, we have the quadratic upper bound:

$$\mathcal{L}(\boldsymbol{\theta}_{t+1}) \leqslant \mathcal{L}(\boldsymbol{\theta}_t) - \eta \langle \nabla \mathcal{L}(\boldsymbol{\theta}_t), \boldsymbol{u}_t \rangle + \frac{\eta^2 H}{2} \|\boldsymbol{u}_t\|_2^2.$$

Taking the expectation at $t$, we have:

$$\mathbb{E}_t \left[ \mathcal{L}(\boldsymbol{\theta}_{t+1}) \right] \leqslant \mathcal{L}(\boldsymbol{\theta}_t) - \eta \mathbb{E}_t \left[ \langle \nabla \mathcal{L}(\boldsymbol{\theta}_t), \boldsymbol{u}_t \rangle \right] + \frac{\eta^2 H}{2} \mathbb{E}_t \left[ \|\boldsymbol{u}_t\|_2^2 \right]$$

$$= \mathcal{L}(\boldsymbol{\theta}_t) - \eta \sum_{i=1}^{d} \mathbb{E}_t \left[ \nabla_i \mathcal{L}(\boldsymbol{\theta}_t) u_{t,i} \right] + \sum_{i=1}^{d} \frac{\eta^2 H}{2} \mathbb{E}_t \left[ u_{t,i}^2 \right].$$

Combine Lemma D.1 with Assumption 4.9, we get

$$\mathbb{E}_t \left[ \nabla_i \mathcal{L}(\boldsymbol{\theta}) u_{t,i} \right] = \mathbb{E}_t \left[ \frac{\nabla_i \mathcal{L}(\boldsymbol{\theta}) \varphi_p(g_{t,i})}{\sqrt{v_{t,i} + \epsilon}} \right] \geqslant \frac{\sigma}{2} \frac{|\nabla_i \mathcal{L}(\boldsymbol{\theta})|^2}{\sqrt{\tilde{v}_{t,i} + \epsilon}} - \frac{2R^p}{\sigma} \mathbb{E} \left[ \frac{\varphi_p^2(g_{t,i})}{v_{t,i} + \epsilon} \right].$$

Using it for each dimension, we have:

$$\mathbb{E}_t \left[ \mathcal{L}(\boldsymbol{\theta}_{t+1}) \right] \leqslant \mathcal{L}(\boldsymbol{\theta}_t) - \frac{\eta \sigma}{2} \frac{|\nabla_i \mathcal{L}(\boldsymbol{\theta}_t)|^2}{\sqrt{\tilde{v}_{t,i} + \epsilon}} + \frac{2\eta R^p}{\sigma} \mathbb{E} \left[ \frac{\varphi_p^2(g_{t,i})}{v_{t,i} + \epsilon} \right] + \sum_{i=1}^{d} \frac{\eta^2 H}{2} \mathbb{E}_t \left[ u_{t,i}^2 \right]$$

$$= \mathcal{L}(\boldsymbol{\theta}_t) - \sum_{i=1}^{d} \frac{\eta \sigma}{2} \frac{|\nabla_i \mathcal{L}(\boldsymbol{\theta}_t)|^2}{\sqrt{\tilde{v}_{t,i} + \epsilon}} + \sum_{i=1}^{d} \left( \frac{2\eta R^p}{\sigma} + \frac{\eta^2 H}{2} \right) \mathbb{E}_t \left[ \frac{\varphi_p^2(g_{t,i})}{v_{t,i} + \epsilon} \right].$$

Noticing $\sqrt{\tilde{v}_{t_i} + \epsilon} \leqslant R^p \sqrt{t}$, we further have:

$$\mathbb{E}_t \left[ \mathcal{L}(\boldsymbol{\theta}_{t+1}) \right] \leqslant \mathcal{L}(\boldsymbol{\theta}_t) - \frac{\eta \sigma}{2} \frac{\|\nabla \mathcal{L}(\boldsymbol{\theta}_t)\|_2^2}{R^p \sqrt{t}} + \sum_{i=1}^{d} \left( \frac{2\eta R^p}{\sigma} + \frac{\eta^2 H}{2} \right) \mathbb{E}_t \left[ \frac{\varphi_p^2(g_{t,i})}{v_{t,i} + \epsilon} \right].$$

Summing the previous inequality for all $0 \leqslant t \leqslant T - 1$, taking the complete expectation, and using $\sqrt{t} \leqslant \sqrt{T}$, we have:

$$\mathbb{E} \left[ \mathcal{L}(\boldsymbol{\theta}_t) \right] \leqslant \mathcal{L}(\boldsymbol{\theta}_0) - \frac{\eta \sigma \sum_{t=1}^{T} \|\nabla \mathcal{L}(\boldsymbol{\theta}_t)\|_2^2}{2 \eta R^p \sqrt{T}} + \sum_{i=1}^{d} \left( \frac{2R^p}{\sigma} + \frac{\eta^2 H}{2} \right) \mathbb{E} \left[ \sum_{t=1}^{T} \frac{\varphi_p^2(g_{t,i})}{v_{t,i} + \epsilon} \right].$$

Then for each dimension, using Lemma D.2 for the sequence $\{(g_{t,i}^p)^2\}_{1\leqslant t\leqslant T}$, we obtain:

$$\mathbb{E}\left[\mathcal{L}(\boldsymbol{\theta}_t)\right]$$

$$\leqslant \mathcal{L}(\boldsymbol{\theta}_0) - \frac{\eta\sigma\sum_{t=1}^{T}\mathbb{E}\|\nabla\mathcal{L}(\boldsymbol{\theta}_t)\|_2^2}{2R^p\sqrt{T}} + \left(\frac{2\eta R^p}{\sigma} + \frac{\eta^2 H}{2}\right) d\,\mathbb{E}\left[\log\left(1 + \frac{1}{\epsilon}\sum_{t=1}^{T}\varphi_p^2(g_{t,i})\right)\right]$$

$$\leqslant \mathcal{L}(\boldsymbol{\theta}_0) - \frac{\eta\sigma\sum_{t=1}^{T}\mathbb{E}\|\nabla\mathcal{L}(\boldsymbol{\theta}_t)\|_2^2}{2R^p\sqrt{T}} + \left(\frac{2\eta R^p}{\sigma} + \frac{\eta^2 H}{2}\right) d\log\left(1 + \frac{R^{2p}}{\epsilon}T\right).$$

This implies:

$$\mathbb{E}\min_{1\leqslant t\leqslant T}\|\nabla\mathcal{L}(\boldsymbol{\theta}_t)\|_2^2 \leqslant \frac{1}{T}\sum_{t=1}^{T}\mathbb{E}\|\nabla\mathcal{L}(\boldsymbol{\theta}_t)\|_2^2$$

$$\leqslant \frac{2R^p}{\sigma\sqrt{T}}\left(\frac{\mathcal{L}(\boldsymbol{\theta}_0) - \mathcal{L}^\star}{\eta} + \left(\frac{2R^p}{\sigma} + \frac{\eta H}{2}\right) d\log\left(1 + \frac{R^{2p}}{\epsilon}T\right)\right)$$

$$\leqslant \frac{R^{p-1}}{\sigma}\frac{2R}{\sqrt{T}}\left(\frac{\mathcal{L}(\boldsymbol{\theta}_0) - \mathcal{L}^\star}{\eta} + \left(2R + \frac{\eta H}{2}\right) d\log\left(1 + \frac{R^2}{\epsilon}T\right)\right)$$

$$= \frac{R^{p-1}}{\sigma}\Big(\text{R.H.S. of (5)}\Big).$$

The last inequality comes from Assumption 4.9 and $R < 1$.

$\square$

# E   STATEMENT

## E.1   LLM USAGE STATEMENT

In this paper, we used LLM to help with writing. The model checked and fixed grammar errors in our text. We also used it to make sentences flow better. The LLM helped improve readability without changing our ideas. We did not use LLM for any other writing tasks. Our use was only for grammar and style improvements.

## E.2   ETHICS STATEMENT

We confirm that this research has been conducted in accordance with the ICLR Code of Ethics . All experiments were performed responsibly, with careful consideration of potential impacts, limitations, and broader societal implications. No part of this work involved practices that violate ethical standards regarding research integrity, fairness, transparency, or the responsible use of computational resources.

## E.3   REPRODUCIBILITY STATEMENT

We believe that all of the experimental results are reproducible in our work. The paper specify comprehensive training and test details (e.g., hyperparameters, how they were chosen, type of optimizer, etc.) necessary to understand the results in Section 3 and Appendix B. Besides, we provide open access to the code in the supplemental material and all data datasets are open-sourced.

