# OpenReview forum: "GradPower: Powering Gradients for Faster Language Model Pre-Training"
_ICLR.cc/2026/Conference — Submitted to ICLR 2026_

### Official Review · Reviewer_Up4f · 2025-10-29

**Soundness:** 2
**Presentation:** 3
**Contribution:** 2
**Rating:** 4
**Confidence:** 3

**Summary:**

The authors introduce the SignPower update, a fairly simple (in a good way) extension to Adam, Muon, etc. that seems to show reasonable, if modest, gains, with occasional outsized gains. The basic idea is that you can raise the gradients to a power (preserving sign) which will, when the power is > 1, accentuate larger gradients and diminish smaller ones. They do experiments with small-ish LLMs and a vision task.

The paper is generally well written. The main experiments are fairly thorough (modulo my main concern below). I'm not much of a theory person, but the proofs seem reasonable if pedestrian. I do worry a major confound is at the root of most of the gains, however.

SignPower introduces two effects that I think need to be distinguished. Because we're clipping gradients, we know |g| < 1 (usually << 1) and so |g|^1.2 will damp gradients. So on the one hand, it decreases all updates ("global damping"); on the other it decreases smaller gradients more than larger gradients ("heavier tails"). The former amounts to an expected dampening of about 6% relative to vanilla Adam.[1]

These effects need to be disentangled.

Footnote:

[1] Assuming ~normally distributed gradients, the adam update rule will be basically Adam(p) = E|g|^p/sqrt(E[|g|^2p]) (let's ignore beta etc)

E[|g|^p] ∝ 2^{p/2} Gamma((p + 1)/2), ignoring constants that cancel. Plugging in we get Adam(1.2)/Adam(1) ≈ 0.94.

Same basic trends hold for the authors' preferred choice of Unif[mu-sigma, mu+sigma] gradients I think, with different numerics. (In my experience, gradients look more gaussian though!)

Disclaimer: I used chatgpt to write the code to compute the various ratios of E[|g|^p] and to check my math.

**Strengths:**

(See also my comments above)

The paper is generally well-written modulo a few typos/grammatical errors that probably come from a deadline rush.

The idea is a relatively small change to Adam (that is part of its appeal) and seems to work reasonably well. They have some theory and experiments to back it up. They even show that it transfers to Muon.

They are clearly not overtuning their method relative to the baseline, which is great practice. They have a few different model classes and two domains.

I could imagine a world where we should be tuning our grad-powers and not our LR (and maybe doing p-schedules instead of LR-schedules). I suspect we aren't in that world, but it would be neat and if we were then this line of work would be very impactful!

**Weaknesses:**

(See also my comments above)

As I said above, I do worry a major confound is at the root of most of the gains. I also worry that, as is usually the case with optimizer papers these days, the scales aren't high enough to be trustworthy, that the gains aren't significant enough that a big lab is going to pick it up (though, maybe due to how easy it is, they will!), and that if they do, it won't scale after proper tuning. This alone shouldn't be taken to mean the paper shouldn't be published: it's the role of academia in this environment to find good ideas that seem to show some promise, and rely on big labs to show that "it matters."

More specifically:

For their main results, the authors do a coarse LR sweep that is structurally quite fair to the baseline (admirably so!). But, given the global damping behavior of the update rule, it seems like one also needs to run each experiment taking into account this effect, especially since the gains are usually relatively small.

For instance, could the authors try running adampower with lr = 1/.94 * adam lr? This would help distinguish the "global damping effect" from the "heavier tails effect." The observation on [254] that adampower makes runs less spiky makes me suspect that the damping is a large part of what's going on. ("Effective but spiky LR" in my experience means lr is slightly too high.) Note that 6% is much less than the resolution of the LR sweep performed.

Section 3.5:

This section also increases my suspicion: Given the implicit LR damping effect,and since the LR was  held constant across all scales other than the smallest, the optimal power would decrease as batch size increased: you want a somewhat higher LR with higher batch size (probably around sqrt(2) per power of 2; see e.g. https://www.cs.princeton.edu/~smalladi/blog/2024/01/22/SDEs-ScalingRules/). Regardless, the differences in figure 6 seem like they could easily be within noise except for the bs=512 with low power and the bs=8192 with high power.

Again, I applaud the authors for doing an LR search for the baseline, but the search domain is fairly coarse. I'd expect the optimal batch size for powers of two batch sizes to differ by sqrt(2) per scale, which is less than the resolution of your search. This may explain why you found similar optimal LRs for those different sizes.

Section 4.1: Same basic deal. Isn't this really an argument for **varying p** (in the same way we vary the LR): following the river-valley analogy, when the SNR is high (on hill), we want lower p (and similarly, high LR), and when it's low, we want higher p (and similarly, low LR). But you aren't varying p within a run! Invoking the river-valley analogy to say that different SNR-regimes require different p's and then using fixed p for all actual runs seems wrong?

**Questions:**

I'd like to see experiments that disentangle the implicit LR damping effect vs the accentuation effect. if you could rerun gradpower with LR = 1/0.94 the adam LR and give me a scaling law (or maybe better, rerun the scaling suite with adam with 0.94* LR) I'd be more convinced. I'd still guess this wouldn't prove itself in the long run, but I'd easily from from a 4 to an 6.

Am I wrong that your experiments in section 4 aren't really supporting your choice of a specific p (but rather support a variable p)?

Minor points:

While gradient clipping is mentioned in passing, it's possibly worth noting the interaction with your method.

045: elemenwise -> elementwise

102: That a linear transform doesn't do anything to adaptive optimizers seems fairly trivial and doesn't merit a half page proof, even in the appendix.

113: The highlight of "one additional line of code" is a bit much

345 : Sentence isn't grammatical

360: noise-dominate -> noise-dominant

Figure 7 description isn't quite grammatical? delete the 'and'?

---

> ### Author Response · Authors · 2025-11-21
> **Response to Reviewer Up4f (1/2)**
>
> We sincerely appreciate the reviewer’s thoughtful and encouraging assessment of our work. We are glad that the clarity of the writing, the simplicity of the proposed update, and the breadth of our experiments came through. It is motivating to see the reviewer acknowledge the potential impact of exploring GradPower. We are grateful for the constructive insights offered throughout the review, which will help us substantially improve the paper.
>
> > **Experiments that disentangle the implicit LR damping effect vs the accentuation effect**.
> > - As I said above, I do worry a major confound is at the root of most of the gains. I also worry that, as is usually the case with optimizer papers these days, the scales aren't high enough to be trustworthy, that the gains aren't significant enough that a big lab is going to pick it up (though, maybe due to how easy it is, they will!), and that if they do, it won't scale after proper tuning. This alone shouldn't be taken to mean the paper shouldn't be published: it's the role of academia in this environment to find good ideas that seem to show some promise, and rely on big labs to show that "it matters." More specifically: For their main results, the authors do a coarse LR sweep that is structurally quite fair to the baseline (admirably so!). But, given the global damping behavior of the update rule, it seems like one also needs to run each experiment taking into account this effect, especially since the gains are usually relatively small.
> > - For instance, could the authors try running adampower with lr = 1/.94 * adam lr? This would help distinguish the "global damping effect" from the "heavier tails effect." The observation on [254] that adampower makes runs less spiky makes me suspect that the damping is a large part of what's going on. ("Effective but spiky LR" in my experience means lr is slightly too high.) Note that 6% is much less than the resolution of the LR sweep performed.
> > - I'd like to see experiments that disentangle the implicit LR damping effect vs the accentuation effect. if you could rerun gradpower with LR = 1/0.94 the adam LR and give me a scaling law (or maybe better, rerun the scaling suite with adam with 0.94* LR) I'd be more convinced. I'd still guess this wouldn't prove itself in the long run, but I'd easily from from a 4 to an 6.
> > - Section 3.5: This section also increases my suspicion: Given the implicit LR damping effect,and since the LR was held constant across all scales other than the smallest, the optimal power would decrease as batch size increased: you want a somewhat higher LR with higher batch size (probably around sqrt(2) per power of 2; see e.g. https://www.cs.princeton.edu/~smalladi/blog/2024/01/22/SDEs-ScalingRules/). Regardless, the differences in figure 6 seem like they could easily be within noise except for the bs=512 with low power and the bs=8192 with high power.
> Again, I applaud the authors for doing an LR search for the baseline, but the search domain is fairly coarse. I'd expect the optimal batch size for powers of two batch sizes to differ by sqrt(2) per scale, which is less than the resolution of your search. This may explain why you found similar optimal LRs for those different sizes.
>
> **Response:** We appreciate the reviewer’s insightful suggestions and detailed analysis. We highly agree that this ablation is valuable for disentangling the "global damping effect" from the "heavier-tails effect", thereby strengthening the empirical interpretation.
> - **New experiments with lr = 0.94 * adam lr.** Following the reviewer's constructive recommendation, we further conducted the proposed scaling law experiment in **0.2B, 0.4B, 1B LLaMA models** under cosine learning rate scheduler. In these runs, we adjust Adam’s learning rate by the reviewer-suggested factor (i.e., lr_new = lr_base * 0.94); full details are provided in Appendix B.5. Due to resource constraints, we have not yet extended this ablation to the MoE or the warmup-stable-decay (wsd) scheduler settings, but we hope that this targeted evaluations address the reviewer’s core concern.
> - **Learning-rate sweep resolution.** We thank the reviewer for the observation regarding the resolution of our learning-rate search. We fully agree that a finer grid (sqrt{2} multiplicative step) would further strengthen the empirical evidence. However, such a sweep requires substantial computational resources, and these experiments are still in progress.

---

> ### Author Response · Authors · 2025-11-21
> **Response to Reviewer Up4f (2/2)**
>
> > Section 4.1: Same basic deal. Isn't this really an argument for **varying p** (in the same way we vary the LR): following the river-valley analogy, when the SNR is high (on hill), we want lower p (and similarly, high LR), and when it's low, we want higher p (and similarly, low LR). But you aren't varying p within a run! Invoking the river-valley analogy to say that different SNR-regimes require different p's and then using fixed p for all actual runs seems wrong?
>
> **Response:** We appreciate the reviewer’s insightful point.
> - We agree that, in principle, adapting $p$ over the course of training could better accommodate different SNR regimes. However, designing a practical $p$ schedule is non-trivial: (i) SNR is difficult to estimate reliably in large-scale training, and (ii) abrupt changes in $p$ may interact unpredictably with momentum.
> - For these reasons, we focus on fixed-$p$ training in this work. We agree that *developing dynamic-$p$ schedules* is an interesting and important direction for future research, and we have clarified this as a future direction in the revised manuscript (Conclusion Section).
>
>
>
> > While gradient clipping is mentioned in passing, it's possibly worth noting the interaction with your method.
>
> **Response:** We thank the reviewer for this insightful comment. Gradient clipping is indeed essential for stabilizing LLM pre-training. In our standard setup, GradPower is applied after gradient clipping. Notably, both orderings induces bounded gradient.
>
> To further assess the interaction between gradient clipping and GradPower, we have conducted **an additional experiment** on LLaMA-0.2B (dense), **in which we switch the order of gradient clipping and the GradPower transformation**, (referred to as AdamPower-II):
> - **Experimental Results:** The terminal loss of AdamPower-II ($p=1.2$) is nearly identical to that of the original AdamPower ($p=1.2$), and the training curves closely match throughout the training process in Appendix B.6.
> - **Understanding:** The effect of the power transformation is relatively consistent for gradient elements with norm less than 1, i.e., suppresses, amplifies. Prior empirical studies [1] suggest that, during stable LLM pre-training, the per-coordinate gradient magnitudes are typically much smaller than 1.0. As a result, changing the order of GradPower and gradient clipping has minimal effect.
>
>
> > Typos.
>
> **Response:** We sincerely thank the reviewer for pointing out these typos; we have corrected them in the revised version.
>
> ------------------
>
> **Reference:**
>
> [1] Huang et al., SPAM: Spike-aware adam with momentum reset for stable LLM training. 2025.

---

> ### Author Response · Authors · 2025-11-27
> **Additional Response to Reviewer Up4f**
>
> Dear Reviewer, thank you again for your constructive feedback! New scaling experiments with `New Adam lr = 0.94 * Old Adam lr` now also cover the **LLaMA on C4, wsd** setting. The results for **LLaMA on C4, cos** (first reponse) and **LLaMA on C4, wsd** (second response) are summarized below, full details are provided in Appendix B.5.
>
> **LLaMA on C4, cos**
> | Model size       | 0.2B                                   | 0.4B                                  | 1B              |
> |------------------|----------------------------------------|---------------------------------------| ---------------------------------------|
> | AdamW            | 3.0006                                 | 2.7889                                | 2.5593          |
> | AdamW (0.94lr)   | 2.9859                                 | 2.7957                                | 2.5601          |
> | AdamWPower (1.2) | **2.9832**                          | **2.7705**                       | **2.5385**|
>
> **LLaMA on C4, wsd**
> | Model size       |    0.4B            | 1B              | 2B              |
> |------------------|----------------------------------------|---------------------------------------| ---------------------------------------|
> | AdamW            | 2.7911          | 2.5645          | 2.4206          |
> | AdamW (0.94lr)    | 2.7917          | 2.5649          | 2.4207          |
> | AdamWPower (1.2)  | **2.7767** | **2.5472** | **2.4028** |

---

### Official Review · Reviewer_cWAA · 2025-10-30

**Soundness:** 3
**Presentation:** 3
**Contribution:** 3
**Rating:** 6
**Confidence:** 2

**Summary:**

This paper proposes GradPower, a simple gradient transformation that applies an element-wise sign-power function to gradients before passing them to any optimizer. Applied to Adam, it claims consistent improvements across LLM pre-training settings, including dense and MoE models, multiple datasets, and learning-rate schedules. The paper also provides theoretical insights into why different powers work in low and high-noise settings.

**Strengths:**

Strength:

* The proposed method is simple and practical in implementation, and can be combined with many advanced optimizer.
* Experiments across multiple LLM scales, datasets, and schedulers demonstrate the effectiveness of the method in accuracy and convergence speed.

**Weaknesses:**

Weaknesses and Question:

* The model scale is limited, the largest one used is 2B. As its mainly focus on pre-training process, it would be better with at least a 7B or 13B baseline to show real scalability, many performance instability issues during training tend to emerge in models larger than 7B.
* Though $p=1.2$ is effective under most settings, but its unclear how sensitive results are to $p$ across broader tasks, or can you provide some results or discussion on adaptive or layer-wise $p$ adjustment?
* The performance gain is borderline, for example in zero-shot, avg accuracy increased 0.5%, and most of increasement comes from one single dataset WINOGRANDE.
* Results limited and small-scale. Need more tasks on standard LLM evals, like including perplexity results or evaluate on benchmarks like MMLU or MT-Bench, which is closer to actual application scenarios.
* Some prior work also tries power-gradient methods, it would be great to include more discussion or quantitative comparison.
* Beside local training loss curve, could you provide more details related to convergence speed? Like GPU hour comparison.

**Questions:**

Please refer to the Weakness part.

---

> ### Author Response · Authors · 2025-11-21
> **Response to Reviewer cWAA**
>
> We appreciate your acknowledgement of simplicity of our method and strength of our experiment. We'd like to clarify following points:
>
> > The model scale is limited, the largest one used is 2B. As its mainly focus on pre-training process, it would be better with at least a 7B or 13B baseline to show real scalability, many performance instability issues during training tend to emerge in models larger than 7B.
>
> **Response:** We thank the reviewer for the question. We fully agree that scalability is an important consideration. We note that training our 2B model under Chinchilla-optimal settings required around 1,000 A100 GPU hours, which is already near the feasible limit for an academic lab. This 2B scale is **comparable to or larger than** those used in prior academic work on optimizer design for LLM pre-training (e.g., [1][2] used up to 1.5B, [3] up to 2B, [4][5][6] up to 1.3B, [7][8] up to 1B). We aim to extend our experiments to larger models as additional computational resources become available.
>
>
> > Though $p=1.2$ is effective under most settings, but its unclear how sensitive results are to across broader tasks, or can you provide some results or discussion on adaptive or layer-wise adjustment?
>
> **Response:** We agree that the choice of $p$ is an important factor. In Section 3.5, we provide a comprehensive discussion and experiments on $p$, showing that the optimal power $p$ decreases as batch size increases, i.e., as the gradient noise level decreases. These observations are also supported by our theoretical analysis.
>
>
> > The performance gain is borderline, for example in zero-shot, avg accuracy increased 0.5%, and most of increasement comes from one single dataset WINOGRANDE. Results limited and small-scale. Need more tasks on standard LLM evals, like including perplexity results or evaluate on benchmarks like MMLU or MT-Bench, which is closer to actual application scenarios.
>
> **Response:** We thank the reviewer for the constructive suggestion.
> - **Downstream tasks.** **New Experiment:** We additionally include a **zero-shot MMLU evaluation on 2B**: GradPower ($p=1.2$) achieves **24.01 vs. 23.37** for the baseline, indicating consistent improvement. We note that MMLU is extremely challenging for these model and data scale, so gains at this scale should be interpreted cautiously. (MT-Bench is not directly applicable to base model pre-training, as it is designed for chat-oriented SFT models.)
> - **Pre-training loss.** Figure 1 in the scaling-law analysis visualizes improvements in convergence speed. The arrow shows how AdamPower saves FLOPS in terms of final loss convergence. As shown in Figure 1, GradPower can save 15%~26% FLOPs comparing to baseline.
>
>
> > Some prior work also tries power-gradient methods, it would be great to include more discussion or quantitative comparison.
>
> **Response:** We thank the reviewer for the suggestion. We have already provided detailed discussion about prior works on power-gradient methods in the Related Work Section on the original manuscript. Prior work primarily focuses on SGD and ImageNet-scale vision models and presents a completely different theoretical framework. To our knowledge, no prior work has demonstrated power-gradient accelerations in the context of modern optimizers such as Adam and Muon or LLM pre-training.
>
>
> > Beside local training loss curve, could you provide more details related to convergence speed? Like GPU hour comparison.
>
> **Response:** We thank the reviewer for the suggestion. Figure 1 in the scaling-law analysis visualizes improvements in convergence speed. The arrow shows how AdamPower saves FLOPS in terms of final loss convergence. As shown in Figure 1, GradPower can save 15%~26% FLOPs comparing to baseline. We believe this serves as a valid GPU hour comparison.
>
> -------------------------
>
> **Reference:**
>
> [1] Liu et al., Sophia: A Scalable Stochastic Second-order Optimizer for Language Model Pre-training. ICLR 2024. \
> [2] Yuan et al., MARS: Unleashing the Power of Variance Reduction for Training Large Models. ICML 2025. \
> [3] Wang et al., The Sharpness Disparity Principle in Transformers for Accelerating Language Model Pre-Training. ICML 2025. \
> [4] Ma et al., SWAN: SGD with Normalization and Whitening Enables Stateless LLM Training. ICML 2025. \
> [5] Pagliardini et al.,The AdEMAMix Optimizer: Better, Faster, Older. ICLR 2025. \
> [6] Marek et al., Small Batch Size Training for Language Models: When Vanilla SGD Works, and Why Gradient Accumulation is Wasteful. NeurIPS 2025. \
> [7] He et al., AlphaDecay: Module-wise Weight Decay for Heavy-Tailed Balancing in LLMs. NeurIPS 2025. \
> [8] Chen et al., Fira: Can We Achieve Full-rank Training of LLMs Under Low-rank Constraint? NeurIPS 2025.

---

### Official Review · Reviewer_yfon · 2025-10-30

**Soundness:** 3
**Presentation:** 4
**Contribution:** 2
**Rating:** 4
**Confidence:** 2

**Summary:**

This paper focus on accelerating large model training by developing a new optimization algorithm. Specifically, the paper studies nonlinear operation on the gradient and proposes signed-power gradient method that powers the magnitude of the gradient for each of its coordinate and keeps its sign. The paper conducts numerical experiments on multiple settings for training large language models.

**Strengths:**

1. Simplicity of the proposed method. The sign power method is light-weighted and is compatible with multiple base optimizers.
2. In the numerical results, the proposed method achieves consistent better performance than thier baselines, and are potentially more stable.
3. The paper provides additional theoretical insights on how the sign power method accelerates the base optimizer (Adam in the 1-d case and Adagrad in high-dimension case)

**Weaknesses:**

1. On the numerical experiment:
    1. Missing reporting the variance of the results. I would expected at least one result reporting the variance of the algorithms with multiple runs (>=5) to show the statistical significance. Given the difference in the loss is only ~1%, it is hard to judge if such difference is comming from the selected seed or due to the proposed algorithm. I understand that for LLM pretraining, multiple run can be time consuming, but it is expected to have at least one set experiment to report the variance.
    2. From the convergence lines, the models are still converging very fast, far before convergence. It is hard to tell how the final model performance look like when the models actually converge.

2. On the hyper-parameter choice
    1. It is hard to see why p=1.2 is a uniformly optimal solution for training large models, given that in Table 2, p=1.2 does not achieve the optimal solution for any of the batch size on Vision model. And is only optimal for batch size 512 and 2048 when training LLaMa 0.2B on C4 (Figure 6). This might suggest that for difference batch size choice (, tasks and models), we need to pick a different p value.

3. On the theoretical result
    1. The restricted assumption: Assumptions 4.6 and 4.9 are quite restrictve, as they both asume that each coordinate of the stochastic gradient should align with the true gradient's direction. This assumption is even stronger than assuming the overall stochastic gradient is aligned with the true gradient. The author should show that under these assumptions, the normal Adagrad is still converging slower than AdagradPower.
    2. In examples 4.7 and 4.10. It is hard to see how 0.99 and 1.01 are derived.
    3. It is inaccurate to say after Theorem 4.8 that the algorithm is 2 times acceleration. It is the rate changed from T^-1/2 to T^-1/(p+1).

**Questions:**

1. Does the theory suggests an optimal choice of p under differen regimes?
2, What is the stochastic gradient is not uniformly bounded, e.g., for a quadratic problem?

---

> ### Author Response · Authors · 2025-11-21
> **Response to Reviewer yfon (1/2)**
>
> We appreciate your acknowledgement of simplicity of the proposed method, consistent improment in numerical experiemnt and additional theoretical insights. We clarify weakness as follows:
>
> > Missing reporting the **variance of the results**. I would expected at least one result reporting the variance of the algorithms with multiple runs (>=5) to show the statistical significance. Given the difference in the loss is only ~1%, it is hard to judge if such difference is comming from the selected seed or due to the proposed algorithm. I understand that for LLM pretraining, multiple run can be time consuming, but it is expected to have at least one set experiment to report the variance.
>
> **Response:** We thank the reviewer for the constructive suggestion. We agree that reporting variance strengthens the robustness of our results. **New experiments:** Accordingly, we **have reported the variance across multiple runs** for the 66M and 200M settings in Appendix B.4.
> We also note that in LLM pre-training, *~1% final loss reduction is practically significant*: as shown in Figure 1, GradPower can save 15%~26% FLOPs comparing to baseline.
>
> > **Fast convergence.** From the convergence lines, the models are still converging very fast, far before convergence. It is hard to tell how the final model performance look like when the models actually converge.
>
> **Response:**
> - We first clarify that our pre-training experiments are conducted with token budgets reaching the **Chinchilla-optimal regime**, where *tokens=$20\times$parameters*. This choice of training steps/tokens follows the common practice in LLM pre-training.
> - Regarding the reviewer’s concern about the visually 'fast convergence', this behavior is an inherent consequence of the `wsd` (Warmup–Stable–Decay) learning rate schedule [1][2]. `wsd` schedule divides training into three distinct phases: an initial warmup phase with increasing learning rate, a stable phase with a constant learning rate, and a final decay phase where the learning rate decreases rapidly. Thus it leads to **a non-traditional loss curve**: *slowly decrease during the stable phase and suddenly drop during the final decay phase.* While the 'fast convergence' may appear dramatic visually, the loss has in fact already *stabilized* by the end of the decay phase.
>
> > **Choice of $p$.** It is hard to see why p=1.2 is a uniformly optimal solution for training large models, given that in Table 2, p=1.2 does not achieve the optimal solution for any of the batch size on Vision model. And is only optimal for batch size 512 and 2048 when training LLaMa 0.2B on C4 (Figure 6). This might suggest that for difference batch size choice (, tasks and models), we need to pick a different p value. Does the theory suggests an optimal choice of p under differen regimes?
>
> **Response:** We thank the reviewer for raising this important question. Although theoretical analysis typically cannot provide quantitatively optimal $p$, both our theory (Section 4) and toy examples (Figure 6, Table 2) reveal **two robust qualitative principles** that consistently govern the optimal choice of $p$ across regimes:
> - Optimal $p$ clearly decreases as batch size increases.
> - (optimal $p$ scaling) **Once batch size and task are fixed, a fixed $p$ performs robustly across model scales.**
>
>
>
> > **The restricted assumption**: Assumptions 4.6 and 4.9 are quite restrictve, as they both asume that each coordinate of the stochastic gradient should align with the true gradient's direction. This assumption is even stronger than assuming the overall stochastic gradient is aligned with the true gradient. The author should show that under these assumptions, the normal Adagrad is still converging slower than AdagradPower.
>
> **Response:** We thank the reviewer for the careful question.
> - Frist, we clarify that Assumptions 4.6 and 4.9 *do not* require each coordinate of *every stochastic gradient* to align with the true gradient. Instead, we only require that each coordinate of the **expected $p$-moment of stochastic gradient** aligns with the true gradient direction.
> - Second, because GradPower operates coordinate-wise, coordinate-level assumptions are essential for the analysis. While we acknowledge that such assumptions may be restrictive, assuming coordinate-wise alignment is a natural and standard step when establishing convergence properties for coordinate-based adaptive methods.
>
>
> > In examples 4.7 and 4.10. It is hard to see how 0.99 and 1.01 are derived.
>
> **Response:** We thank the reviewer for the careful question. The constants 0.99 and 1.01 are illustrative and can be replaced by any constants sufficiently close to 1 (e.g., 0.999 or 1.001). For example, the 0.99 arises from the assumption $\sigma \ll |\nabla_i\ell|$, which implies $1+o(\sigma/|\nabla_i\ell|) \ge 0.99$. Similarly, 1.01 follows from the assumption $\sigma_i \gg |\nabla_i\ell|$, which yields $(|\nabla_i\ell| + \sigma_i)^{p-1} \le 1.01\sigma_i^{p-1}$.

---

> ### Author Response · Authors · 2025-11-21
> **Response to Reviewer yfon (2/2)**
>
> > It is inaccurate to say after Theorem 4.8 that the algorithm is 2 times acceleration. It is the rate changed from T^-1/2 to T^-1/(p+1).
>
> **Response:**  To clarify, Theorem 4.8 *does not* claim that the algorithm is 2 times acceleration. Instead, our Line 447-338 shows that the convergence improves by a factor of $2/(p+1)$. In particular, this factor approaches 2 times in the limit as $p\to 0$.
>
>
> > What is the stochastic gradient is not uniformly bounded, e.g., for a quadratic problem?
>
> **Response:** Thank you for the thoughtful question. We first note that the assumption of uniformly bounded stochastic gradients is standard in stochastic optimization theory (for adaptive optimizers), as used, for example in [3]. Empirically, this assumption also holds in large-scale language-model pre-training [4]. Nevertheless, we agree that extending the theory to settings with unbounded stochastic gradients is an important and interesting direction, and we leave this for future work.
>
> ---------------------------
>
> **Reference:**
>
> [1] Hu et al., MiniCPM: Unveiling the Potential of Small Language Models with Scalable Training Strategies, COLM 2024. \
> [2] Wen et al., Understanding Warmup-Stable-Decay Learning Rates, ICLR 2025. \
> [3] Défossez et al., A simple convergence proof of adam and adagrad, TMLR 2022. \
> [4] Huang et al., Spam: Spikeaware adam with momentum reset for stable llm training, ICML 2025.

---

> > ### Comment · Reviewer_yfon · 2025-11-25
> >
> > Thank the authors for their feedback. The additional numerical results demonstrate a consistent improvement across multiple runs.
> >
> > I don't have further questions or concerns.

---

> > > ### Author Response · Authors · 2025-11-26
> > >
> > > We would like to reiterate our sincere gratitude for your valuable recommendation and positive feedback. We are pleased to have addressed your concerns and appreciate your decision to raise the score. Thank you!

---

### Official Review · Reviewer_uCom · 2025-11-04

**Soundness:** 3
**Presentation:** 3
**Contribution:** 2
**Rating:** 4
**Confidence:** 4

**Summary:**

The paper propose GradPower, a new gradient-transformetion technique that can orthogonally combined with AdamW and Muon to reach faster convergence.

**Strengths:**

The topic of boosting optimizer performance is important. The writing is mostly clear. The experiments are sound.

**Weaknesses:**

The motivation is unclear, as I elaborate as follows.

**Questions:**

**Major concern:** I think the motivation of the proposed GradPower operator is quite unclear.

If I understand correctly, the motivation is to "line 68: amplifies these (flat) directions". I do not understand how to achieve this goal with GradPower.  In optimization theory, flat direction usually refers to the eigenvectors of the Hessian associated with small eigenvalues.  "Amplify these (flat) directions" requires first finding the targeted Hessian eigenvectors, and then amplifying the update along them.

In principle, in order to  "amplify these (flat) directions" or "take larger steps in the flat directions", one needs the following procedure:

Step 1. Rotate the gradient vector into a coordinate system under the eigenbasis of the Hessian, in which the Hessian will become diagonal and the eigenvectors become the standard basis.

Step 2.  Amplify the update on the small-diagonal component in Hessian. This equivalently amplifies the update along the Hessian eigenvector.

Step 3. Rotate back to the original coordinate system and update the optimization variable.

Now, how does GradPower operator ensure all these？GradPower simply takes the sign and power over the gradient component.   This indeed amplifies the small gradient components, but this does not mean we "take larger steps in the flat directions".  I do not know how GradPower helps achieve this without the rotation in Steps 1 and 3. Why is such a design reasonable?  Is there any theory supporting that we can skip the rotation steps? Is there any implicit approximation of the rotation? The current theory in Section 4.1 does not help address this concern.

With that being said, it is possible that I misunderstood the meaning of  "amplifies these (flat) directions". Please explain what it means if this is the case.

---

> ### Author Response · Authors · 2025-11-21
> **Response to Reviewer uCom**
>
> We appreciate your acknowledgement of the importance of boosting optimizer performance, clear writing and sound experiments. *Below we clarify what “amplifying flat directions’’ means in our paper, and why it does not require explicitly computing Hessian eigenvectors.*
>
>
> > Major concern: I think the motivation of the proposed GradPower operator is quite unclear.
> If I understand correctly, the motivation is to "line 68: amplifies these (flat) directions". I do not understand how to achieve this goal with GradPower. In optimization theory, flat direction usually refers to the eigenvectors of the Hessian associated with small eigenvalues. "Amplify these (flat) directions" requires first finding the targeted Hessian eigenvectors, and then amplifying the update along them.
> In principle, in order to "amplify these (flat) directions" or "take larger steps in the flat directions", one needs the following procedure:
> Step 1. Rotate the gradient vector into a coordinate system under the eigenbasis of the Hessian, in which the Hessian will become diagonal and the eigenvectors become the standard basis.
> Step 2. Amplify the update on the small-diagonal component in Hessian. This equivalently amplifies the update along the Hessian eigenvector.
> Step 3. Rotate back to the original coordinate system and update the optimization variable.
> Now, how does GradPower operator ensure all these？GradPower simply takes the sign and power over the gradient component. This indeed amplifies the small gradient components, but this does not mean we "take larger steps in the flat directions". I do not know how GradPower helps achieve this without the rotation in Steps 1 and 3. Why is such a design reasonable? Is there any theory supporting that we can skip the rotation steps? Is there any implicit approximation of the rotation? The current theory in Section 4.1 does not help address this concern.
> With that being said, it is possible that I misunderstood the meaning of "amplifies these (flat) directions". Please explain what it means if this is the case.
>
> **Response:** We thank the reviewer for raising this insightful question.
> - **Clarification of flat directions.** We agree that in optimization theory, flat directions refer to Hessian eigenvectors associated with small eigenvalues. However, our usage is *approximate* and follows a line of prior work showing that the **anisotropy of gradient noise** closely reflects the Hessian’s curvature structure. These works [1][2][3][4] establish that **directions with small Hessian curvature exhibit low gradient-noise variance, while directions with large Hessian curvature exhibit high gradient-noise variance.** Consequently, flat directions approximately correspond to low-noise directions, and sharp directions to high-noise directions. We've added above explanation to Related Work in the revised version.
> - **Our method.** Under this correspondence, GradPower does not require explicit computation of the Hessian eigenbasis, and therefore Steps 1–3 described by the reviewer (rotation $\to$ amplification $\to$ inverse rotation) are unnecessary. GradPower operates directly on the **per-coordinate stochastic gradient**, whose anisotropy already reflects Hessian anisotropy through the noise structure. As shown in Section 4, *GradPower enlarges the effective update magnitude in low-noise directions and suppresses it in high-noise ones*, thereby **implicitly amplifying flat directions** without requiring an *explicit* Hessian-eigendecomposition. In this sense, the 'rotation' is not skipped but rather *implicitly encoded in the anisotropy of stochastic gradients*.
>
>
> --------------------------------
>
> **Reference:**
>
> [1] Zhu et al. The Anisotropic Noise in Stochastic Gradient Descent: Its Behavior of Escaping from Sharp Minima and Regularization Effects, ICML 2019.\
> [2] Wu et al. On the noisy gradient descent that generalizes as sgd. International Conference on Machine Learning. ICML 2020.\
> [3] Mori et al. Power-law escape rate of SGD. International Conference on Machine Learning. ICML 2022.\
> [4] Wu, Wang, and Su. The alignment property of SGD noise and how it helps select flat minima: A stability analysis. NeurIPS 2022.

---

> > ### Comment · Reviewer_uCom · 2025-11-24
> > **A follow-up question**
> >
> > Thanks for the rebuttal!
> > In the rebuttal, the authors mentioned that "directions with small Hessian curvature exhibit low gradient-noise variance", and "flat directions approximately correspond to low-noise directions". Several papers are cited to support the claim.
> >
> > Though I haven't carefully checked the references, I can live with this claim and assume it is true (for now).  However, there is still a gap between this claim and the proposed method: the proposed method aims to amplify "flat directions" by amplifying the small "small stochastic gradient entry", but not "low gradient-noise variance". Why does "small stochastic gradient entry" imply "low gradient-noise variance"? To me, these two are not directly related.  I do not quite follow the logic here. Please provide more explanation or evidence.

---

> ### Author Response · Authors · 2025-11-26
> **Response to the follow-up question**
>
> Thank you for the thoughtful and constructive question! We appreciate your agreement on the relationship between Hessian structure and noise anisotropy: "directions with small Hessian curvature exhibit low gradient-noise variance", and "flat directions approximately correspond to low-noise directions". We now address your follow-up question:
>
> Our argument relies on the observation that LLM pre-training typically operates in a **noise-dominated regime**, where *the stochastic gradient is largely governed by gradient noise*. In this regime, coordinates with smaller stochastic-gradient magnitude naturally correspond to coordinates with smaller gradient-noise variance. Below we provide additional explanation and supporting evidence:
>
> - **Intuition**. In large-scale LLM pre-training, the dataset is massive, and each mini-batch constitutes only a tiny fraction of it. As a result, the stochastic gradient is dominated by sampling noise. Therefore, relative differences across coordinates in stochastic gradient $g$ primarily arise from differences in the noise scale $\xi$, rather than differences in the true gradient $\nabla L$. (Note: $\xi=g-\nabla L$).
>
> - **Empirical support**. Prior empirical studies in conventional deep-learning settings (e.g., ImageNet) show that as training progresses, *the gradient noise constitutes the dominant component of the stochastic gradient*, and the dynamics occurs in the noise-dominated regime [1][2]. For LLMs, direct empirical verification is challenging because computing the full-batch gradient is infeasible, but the above intuition suggests that LLM training should also lie in this noise-dominated regime.
>
>
> --------------------------
>
> **Reference:**
>
> [1] McCandlish et al., An Empirical Model of Large-Batch Training 2018.
>
> [2] Sun et al., Unleashing the Power of Gradient Signal-to-Noise Ratio for Zero-Shot NAS. ICCV 2023.

---

### Author Response · Authors · 2025-12-03
**Global Response**

Dear AC and reviewers,

We appreciate the reviewers' efforts throughout the evaluation process. We are encouraged by their recognition of our work -- its theoretical back (Reviewer **yfon**, **Up4f**), sound experiment (Reviewer **uCom**) with consistent improvement across multiple LLM scales, datasets, and schedulers (Reviewer **yfon**, **cWAA**), and simpliticy of our method (Reviewer **yfon**, **cWAA**, and **Up4f**).

We have responded to each reviewer in detail, below we provide a concise summary.

- **Reviewer uCom.** *(rating: 4, response: major concern resolved, raising a follow-up question)* We address the main concern about the meaning of "amplifies these (flat) directions" by showing a line of prior work demonstrating how the anisotropy of gradient noise reflects Hessian’s curvature structure, which **has been acknowledged by the reviewer**. We also address remaining concern by clarifing the reviewer's recognition "GradPower aims to amplify flat directions by amplifying the small small stochastic gradient entry".
- **Reviewer yfon.** *(rating: 4 to 6, response: concern resolved)* We address the main concern about experimental variance by adding multi-seed experiments in Appendix B.4. We also clarify the rest of concerns about visually 'fast convergence' resulted by wsd lr schedule and other details in the paper. **The reviewer has acknowledged these clarifications and raised the score from 4 to 6 (though no longer visible).**
- **Reviewer cWAA.** *(rating: 6, response: no)* We clarify that our 2B Chinchilla scale is **comparable to or larger than** those used in prior academic work on optimizer design for LLM pre-training (e.g., [1][2] used up to 1.5B, [3] up to 2B, [4][5][6] up to 1.3B, [7][8] up to 1B). We also clarify rest concerns about the choice of $p$, downstream gains, related work and convergence speed comparison.
- **Reviewer Up4f.** *(rating: 4, response: no)* We address the reviewer's main concern by rerunning experiments that "disentangle the implicit LR damping effect vs the accentuation effect"—"the scaling suite with adam with 0.94*LR"—in Appendix B.5, which **the reviewer clearly indicated could easily raise from from a 4 to an 6**. We also address the reviewer's remaining concerns, adding experiments discussing the interaction with gradient clipping and GradPower in Appendix B.6.

Due to special security incident, some reviewers have not responded yet, which we totally understand. We sincerely thank AC and  reviewers for their effort and work!

**All additional experimental results are included in the revised paper.**

Best regards,

Authors of Submission 17995

-----------------------------------------

**Reference:**

[1] Liu et al., Sophia: A Scalable Stochastic Second-order Optimizer for Language Model Pre-training. ICLR 2024. \
[2] Yuan et al., MARS: Unleashing the Power of Variance Reduction for Training Large Models. ICML 2025. \
[3] Wang et al., The Sharpness Disparity Principle in Transformers for Accelerating Language Model Pre-Training. ICML 2025. \
[4] Ma et al., SWAN: SGD with Normalization and Whitening Enables Stateless LLM Training. ICML 2025. \
[5] Pagliardini et al.,The AdEMAMix Optimizer: Better, Faster, Older. ICLR 2025. \
[6] Marek et al., Small Batch Size Training for Language Models: When Vanilla SGD Works, and Why Gradient Accumulation is Wasteful. NeurIPS 2025. \
[7] He et al., AlphaDecay: Module-wise Weight Decay for Heavy-Tailed Balancing in LLMs. NeurIPS 2025. \
[8] Chen et al., Fira: Can We Achieve Full-rank Training of LLMs Under Low-rank Constraint? NeurIPS 2025.

---

### Meta-Review · Area_Chair_XqkZ · 2026-01-06

**Summary:**

This paper introduces GradPower, an element-wise power transformation of gradients that can be plugged into existing optimizers, and reports consistent empirical gains for large-scale language model pre-training, supported by extensive experiments and accompanying theory . The paper is clearly written and technically solid, with broad experimental coverage.

However, I am leaning toward **rejection** due to two main unresolved concerns raised by the reviewers. First, **Reviewer Up4f** points out a potentially major **confound with gradient clipping**. Since clipping is always enabled in the experiments, it remains unclear whether the gains arise from the proposed transformation itself or from altered effective clipping behavior. This interaction needs to be addressed more thoroughly, both theoretically and through targeted ablations.

Second, **Reviewer yfon’s concerns about the theoretical grounding** are only partially resolved. While the paper contains substantial theory, the connection between the analyzed settings and practical Adam-based LLM pre-training remains indirect, and some key assumptions are not fully clarified.

Overall, the idea is interesting and potentially impactful, but the remaining methodological and theoretical issues prevent a positive recommendation at this time.

**Reviewer Concerns:**

See above.

**Reviewer Scores:**

Based on the discussion and rebuttal, I believe that at least one reviewer would have modestly increased their score to reflect improved clarity and additional explanations provided by the authors. However, reviewers who raised fundamental concerns, particularly regarding the interaction with gradient clipping and the strength of the theoretical justification, would likely have maintained their original scores or weakly supporting acceptance, as these core issues were not fully resolved.

---

### Decision · Program_Chairs · 2026-01-26

Reject